# Nuclear fascin regulates cancer cell survival

Campbell D Lawson[1], Samantha Peel[2], Asier Jayo[1†], Adam Corrigan[2], Preeti Iyer[3], Mabel Baxter Dalrymple[1], Richard J Marsh[1], Susan Cox[1], Isabel Van Audenhove[4], Jan Gettemans[4], Maddy Parsons[1]*

[1]Randall Centre for Cell and Molecular Biophysics, King's College London, Guy's Campus, London, United Kingdom; [2]Discovery Sciences, R&D, AstraZeneca (United Kingdom), Cambridge, United Kingdom; [3]Molecular AI, Discovery Sciences, R&D, AstraZeneca (Sweden), Mölndal, Sweden; [4]Department of Biomolecular Medicine, Faculty of Medicine and Health Sciences, Ghent University, Ghent, Belgium

**Abstract** Fascin is an important regulator of F-actin bundling leading to enhanced filopodia assembly. Fascin is also overexpressed in most solid tumours where it supports invasion through control of F-actin structures at the periphery and nuclear envelope. Recently, fascin has been identified in the nucleus of a broad range of cell types but the contributions of nuclear fascin to cancer cell behaviour remain unknown. Here, we demonstrate that fascin bundles F-actin within the nucleus to support chromatin organisation and efficient DDR. Fascin associates directly with phosphorylated Histone H3 leading to regulated levels of nuclear fascin to support these phenotypes. Forcing nuclear fascin accumulation through the expression of nuclear-targeted fascin-specific nanobodies or inhibition of Histone H3 kinases results in enhanced and sustained nuclear F-actin bundling leading to reduced invasion, viability, and nuclear fascin-specific/driven apoptosis. These findings represent an additional important route through which fascin can support tumourigenesis and provide insight into potential pathways for targeted fascin-dependent cancer cell killing.

*For correspondence:
maddy.parsons@kcl.ac.uk

Present address: [†]Faculty of Education, UNIR International University of la Rioja, La Rioja, Spain

## Editor's evaluation

This paper significantly extends previous work suggesting a role for fascin in the nucleus, with the authors concluding that it contributes to multiple aspects of cancer cell regulation and behaviour. The authors used a combination of methodologies, including biochemistry, excellent cell and actin imaging, controlled nanobody-mediated targeting of fascin to the nucleus (when endogenous fascin has been suppressed), proteomics, cancer cell biological assays, and high-content phenotypic screening to identify potential regulators, and function, of nuclear fascin, and the consequences of maintaining too high levels of fascin in the nucleus. Fascin is an important protein in cancer cell behaviour, and this work provides novel information on dynamic active transport in and out of the nucleus, on its role in nuclear actin bundling, its binding to histone-H3, and its contribution to the DNA damage response (DDR; monitored by gammaH2AX foci accumulation), chromatin compaction, cell migration, and cancer cell invasion in an in vitro assay; moreover, dynamic regulation of nuclear fascin is important because too much triggers apoptosis.

## Introduction

Fascin is an F-actin-binding protein that promotes the parallel bundling of actin filaments (*Vignjevic et al., 2006a*). These actin bundles can take the form of filopodia (extending beyond the plasma membrane), or microspikes (within lamellae of migrating cells or neuronal growth cones) and are

involved in numerous biological processes and pathologies (*Hashimoto et al., 2011*; *Mattila and Lappalainen, 2008*; *Wood and Martin, 2002*), including promoting cell migration (*Adams, 2004*) and during embryonic development (*Zanet et al., 2009*). Importantly, fascin expression is very low or absent in normal adult epithelia, and a dramatic up-regulation at both gene and protein levels has been reported in the majority of human carcinomas studied to date (*Hashimoto et al., 2005*). Thus, fascin is emerging as a key prognostic marker and a potential therapeutic target for metastatic disease. The migration and invasion of carcinoma cells are highly coordinated processes that depend largely on both alterations to cell-cell and cell-extracellular matrix (ECM) adhesion and to the signalling events responsible for organisation of the actin cytoskeleton (*Guo and Giancotti, 2004*). Cancer cells migrating in 3D ECM and in tissue assemble membrane protrusions and ECM-degrading adhesions termed invadopodia to enable tunnelling through matrix (*Condeelis et al., 2005*; *Friedl and Wolf, 2003*; *Li et al., 2010*). Across a range of cell types, loss of fascin function results in reduced migration and invasion in vivo (*Chen et al., 2010*; *Hashimoto et al., 2007*; *Jayo et al., 2012*; *Kim et al., 2009*; *Zanet et al., 2012*). Fascin control of filopodia is in part coordinated spatially by formins that nucleate assembly of F-actin structures (*Pfisterer et al., 2020*). Fascin expression promotes increased cytoskeletal dynamics, migratory capacity, and thereby the potential for metastasis.

Fascin is made up of four β-trefoil (βt) repeats separated by short flexible linker regions. Biochemical and modelling data have identified N- and C-terminal actin-binding sites in fascin that enable efficient bundling activity (*Jansen et al., 2011*). We and other groups have previously demonstrated that protein kinase C (α/γ) phosphorylates the highly conserved S39 site within βt1 of fascin and this negatively regulates actin bundling, cell protrusion assembly, and migration (*Adams et al., 1999*; *Anilkumar et al., 2003*; *Hashimoto et al., 2007*; *Parsons and Adams, 2008*; *Vignjevic et al., 2006b*). We also identified S274 as a second conserved phosphorylated site that lies within the C-terminal actin-binding site that contributes to actin bundling and plays an additional role in the control of microtubule dynamics (*Villari et al., 2015*; *Zanet et al., 2012*). Furthermore, we have shown that fascin is localised to the nuclear envelope (NE) where it binds directly to the nucleo-cytoplasmic linker protein nesprin-2 and regulates nuclear plasticity leading to enhanced cell invasion (*Jayo et al., 2016*). Additionally, we have demonstrated that fascin is located within the nucleus where it can bundle F-actin (*Groen et al., 2015*), but the role for nuclear-localised fascin and the potential impact on cancer cell behaviour remain unknown. Moreover, the general and distinct mechanisms that control these dynamic changes in fascin-binding partners, localisation, and function remain poorly understood.

The nucleus contains high levels of actin, both monomeric and filamentous (*Pederson, 2008*), with actin filaments being shorter in length than those found in the cytoplasm. Our data have shown that fascin is required for endogenous nuclear actin bundles to form, and depletion of fascin from *Drosophila* nurse cells increases the size and number of nucleoli, suggesting a role in maintaining nuclear actin organisation and compartments (*Groen et al., 2015*). Several other actin-binding proteins, whose functions have been extensively investigated in the cytoplasm, are also present in the nucleus, including cofilin (suggested to be important for actin monomer import and accumulation in the nucleus) and profilin (partly responsible for actin nuclear export; *Falahzadeh et al., 2015*). Transient actin filaments, detected in nuclei upon serum stimulation or cell spreading, are reported to regulate MAL (MRTF or MKL1) transcription factor activation (*Baarlink et al., 2013*; *Plessner et al., 2015*). However, their effects on general transcription and chromatin remodelling are poorly understood. As high amounts of fascin are only present in certain stages of *Drosophila* follicle development and in growing human cancer cells (*Groen et al., 2015*), it is plausible that fascin import and export into and out of the nucleus is also tightly regulated depending on cell cycle, stress, or environmental influences. Thus, the identification of regulatory signals controlling dynamic movement of fascin within different subcellular compartments represents an important and unexplored area that may lead to new therapeutic avenues in cancer treatment.

In the present study, we demonstrate that fascin is actively transported into the nucleus where it supports post-mitotic F-actin bundling, efficient DNA damage response (DDR), and cancer cell survival. Using high-content phenotypic imaging, we identify key upstream regulatory pathways that promote nuclear fascin and F-actin levels and demonstrate that forcing sustained nuclear fascin reduces cell invasion and promotes apoptosis. We further uncover roles for upstream kinases controlling Histone H3 phosphorylation in mediating fascin-dependent nuclear-DNA tethering and in controlling subsequent

apoptotic responses. Our findings highlight the importance of dynamic nuclear fascin recruitment for tumour cell survival and provide new targets to explore for targeted cancer cell killing.

## Results

### Nuclear fascin contributes to efficient nuclear F-actin bundling

We have previously reported that fascin is localised to the nucleus in a range of cell types in vitro and in vivo (*Groen et al., 2015*). We further verified localisation of endogenous fascin to the nucleus in two human cancer cell lines, HeLa and MDA-MB-231 (used throughout this study), and validated specificity of staining using stable fascin knockdown (KD) cells in each case (*Figure 1—figure supplement 1A*). We then analysed the sequence of fascin to determine whether any potential nuclear localisation (NLS) or nuclear export signal (NES) sequences existed that may mediate this nuclear transport. Our in silico analysis identified putative bipartite NLS and NES sequences (*Figure 1—figure supplement 1B*). We mutated these sequences and performed biochemical fractionation analysis that demonstrated a clear nuclear accumulation of the NES mutant but not the NLS mutant (*Figure 1A*; *Figure 1—source data 1*), indicating that fascin is actively transported into and out of the nucleus. Moreover, treatment of cells with leptomycin B, which inhibits nuclear export through blocking exportin 1, also resulted in accumulation of nuclear fascin (*Figure 1—figure supplement 1C*). It is notable that another group have identified the same NLS in fascin but suggested that removal of this sequence did not alter F-actin binding (*Saad et al., 2016*). In our hands, however, deletion or mutation of any of the NLS residues led to loss of F-actin binding (not shown), which is perhaps unsurprising given the very close proximity to the crucial S39 residue required for actin bundling. While this proximity to S39 is interesting in terms of fascin regulation, we chose not to use the NLS mutants for further functional analysis due to our concerns regarding loss of function.

To investigate nuclear fascin effects on F-actin assembly, we depleted fascin from HeLa cells and stained with an N-terminal anti-actin antibody (AC15) that recognises nuclear actin (*Miyamoto et al., 2011*). Images demonstrated a striking loss of endogenous nuclear F-actin in fascin-depleted cells compared to controls (*Figure 1—figure supplement 1D*). We next sought to determine whether nuclear fascin induces nuclear F-actin bundles. Expression of actin fused to an NLS and FLAG epitope tag (actin-NLS) revealed long, crosslinked F-actin bundles decorated with endogenous fascin in control cells (*Figure 1—figure supplement 1E*). Moreover, GFP-fascin expressed in fascin KD HeLa cells also colocalised with nuclear actin-NLS-tagged filaments and assembly of these filaments was markedly reduced in fascin KD cells within the same field of view (*Figure 1—figure supplement 1F*). Further confirmation of the requirement for fascin-dependent nuclear F-actin assembly was demonstrated by data showing that S39A fascin, which constitutively bundles F-actin, could induce nuclear F-actin bundles, but the non-bundling mutant S39D could not (*Figure 1B*). Ratios of WT, S39A, and S39D fascin in nuclear vs. cytoplasmic fractions were very similar (*Figure 1A*), indicating the differences in nuclear F-actin assembly were not due to variation in nuclear abundance of these proteins. These data demonstrate that fascin is required for efficient nuclear F-actin bundling and this activity requires the N-terminal actin-binding site.

We next assessed how fascin contributed to dynamic assembly of nuclear F-actin. Previous studies have shown nuclear F-actin assembles in nuclei in re-spreading, post-mitotic cells to support nuclear expansion in early G1 (*Baarlink et al., 2017*). To assess whether fascin play a role in this process, we expressed a nanobody that recognises F-actin tagged with both an NLS and iRFP (iRFP-nAC) in HeLa cells stably depleted of fascin and re-expressing GFP-fascin or GFP only. Live imaging of GFP-fascin cells identified that fascin transiently localised to the nucleus following cytokinesis. In agreement with previous reports, nuclear F-actin was also found to transiently assemble post-mitosis in these cells. However, F-actin assembly was visibly impaired in fascin-depleted cells (*Figure 1C*). Quantification confirmed a significant reduction in the duration of nuclear F-actin assembly post-mitosis in fascin KD cells (*Figure 1D*). To quantify F-actin structure, we developed an automated analysis pipeline that enables segmentation and analysis of the structure of nuclear F-actin marked by iRFP-nAC. Higher values from this automated analysis indicate more highly organised and contiguous nuclear F-actin; lower values indicate short or no nuclear actin filaments. Quantification of iRFP-nAC from post-mitotic cells revealed a significant reduction in nuclear F-actin organisation in fascin-depleted cells

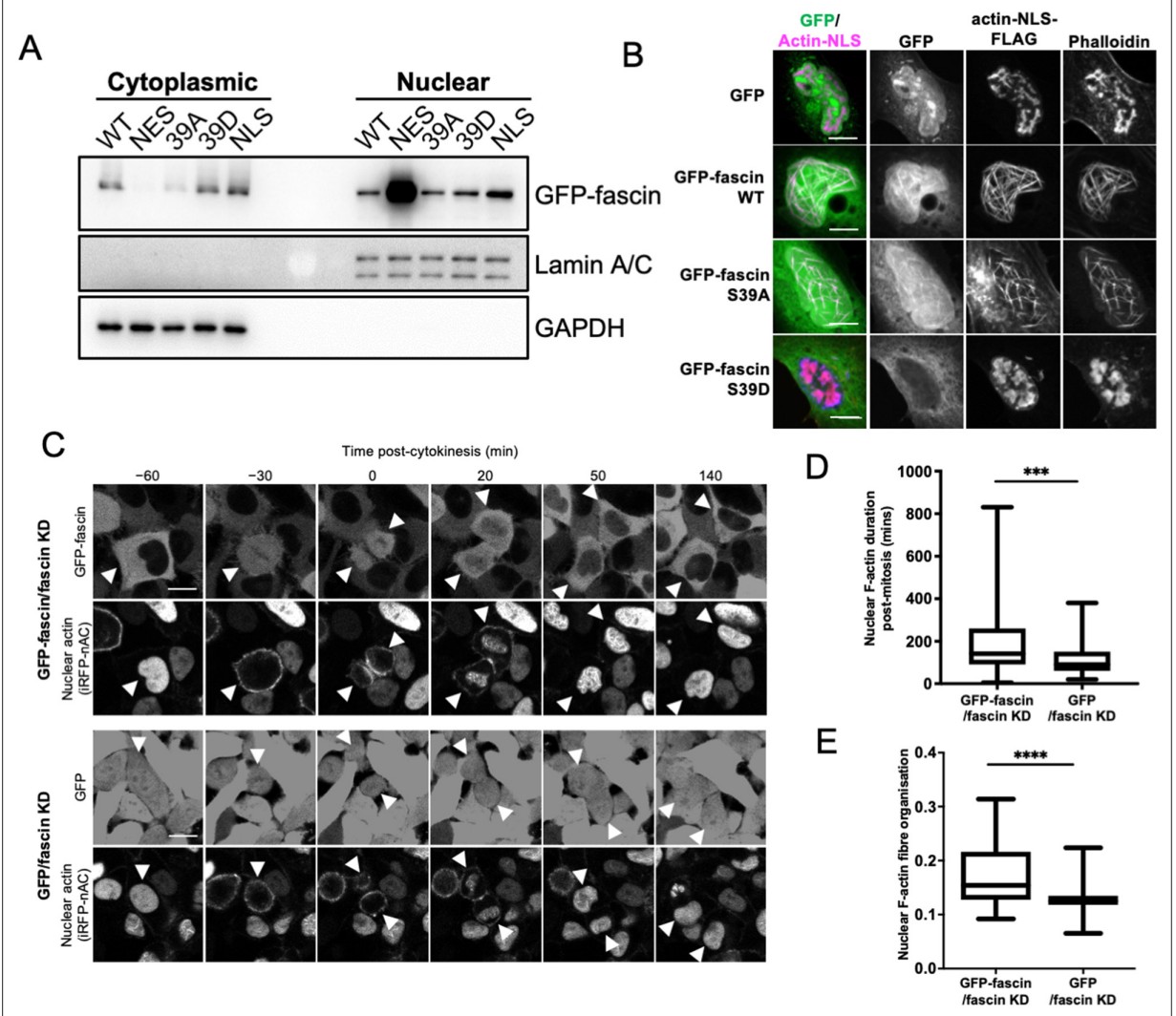

**Figure 1.** Nuclear fascin contributes to F-actin bundling. (**A**) Representative western blot of fascin knockdown (KD) HeLa cells expressing specified GFP-fascin constructs subjected to biochemical fractionation. Nuclear and cytoplasmic compartments probed for GFP-fascin (80 kDa), Lamin A/C (69/62 kda) and GAPDH (36 kDa). Representative of three independent experiments. (**B**) Representative confocal images of nuclei of fascin KD HeLa cells co-expressing specified GFP-fascin constructs (green) and actin-NLS-FLAG construct, fixed and stained for FLAG (magenta) and F-actin (phalloidin). Scale bars are 10 µm. (**C**) Representative stills from time-lapse confocal movies of fascin KD HeLa cells co-expressing GFP or GFP-fascin (top panels) and iRFP-nAC nuclear F-actin probe (bottom panels) pre- and post-cytokinesis. Arrowheads point to dividing or daughter cells. Scale bars are 10 µm. (**D**) Quantification of duration of nuclear F-actin filaments in cells as in (**C**). (**E**) Organisation of nuclear F-actin in synchronised cells, 10 hr after release. For (**D**) and (**E**), N=89–100 cells/condition, pooled from three independent experiments. Graphs shows min/max and mean of dataset. ***=p < 0.001, ****=p < 0.0001.

The online version of this article includes the following source data and figure supplement(s) for figure 1:

**Source data 1.** *Figure 1A* full western blots.

**Figure supplement 1.** Fascin localises to the nucleus and colocalises with nuclear F-actin.

(*Figure 1E*). This data collectively demonstrates a requirement for nuclear fascin to support efficient nuclear F-actin assembly.

## Nuclear fascin directly associates with Histone H3

To explore potential nuclear fascin-binding partners other than F-actin, we adapted previously characterised fascin-specific GFP- or mCherry-tagged expressible nanobodies (*Van Audenhove et al., 2014*) by inserting NLS or NES sequences, which enabled movement of endogenous fascin to the nucleus or cytoplasm, respectively. GFP or mCherry alone and GFP- or mCherry-fascin nanobody

(Nb2) were used as controls. Notably, Nb2 has previously been shown not to affect fascin function when expressed in cells (*Van Audenhove et al., 2014*). We verified successful expression and expected relocation of endogenous fascin in cells expressing NLS and NES mCherry-Nb2 plasmids, with no change seen in endogenous fascin localisation in controls (*Figure 2A*). Biochemical fractionation experiments further confirmed the enrichment of nuclear fascin in cells expressing Nb2-NLS (*Figure 2B and C*; *Figure 2—source data 1*). Cells expressing GFP-Nb2 plasmids were subjected to GFP-Trap to immunoprecipitate Nb2, and selected associated complexes were then identified by proteomic analysis. The resulting data were analysed and prioritised based on those hits with highest numbers of unique peptides identified in Nb2-NLS but not Nb2-NES complexes. Data revealed actin as the top hit across all Nb2 samples, with Histones H3 and H4 the most highly represented proteins in the Nb2-NLS samples only (*Figure 2—source data 2*). We chose to pursue Histone H3 for further validation to determine whether this protein is a nuclear-binding partner for fascin. Repeat GFP-Trap experiments verified the association of Histone H3 with Nb2-NLS but not Nb2-NES (*Figure 2D*; *Figure 2—source data 3*) and co-staining of cells for endogenous proteins revealed partial colocalisation within the nucleus (*Figure 2E*). To determine whether the fascin-Histone H3 association was direct, we co-expressed GFP-Histone H3 and mCherry-Nb2-NLS in cells and analysed binding using fluorescence resonance energy transfer (FRET). A range of FRET efficiencies were seen within a population of cells (*Figure 2F and G*), demonstrating that fascin and Histone H3 can directly associate and suggesting Histone H3 may act as a nuclear tether for fascin, contributing to fascin-dependent functions within this compartment. To further explore the relationship between fascin-F-actin bundling and fascin-Histone H3 binding, we co-expressed RFP-WT, S39A, and S39D fascin with GFP-Nb2-NLS followed by GFP-Trap to enrich for nuclear fascin species. Subsequent probing of these complexes revealed a loss of Histone H3 binding to S39A fascin compared to WT or S39D fascin, indicating that fascin-dependent F-actin bundling reduces the interaction of fascin with Histone H3 (*Figure 2H*; *Figure 2—source data 4*). Taken together these findings demonstrate that nuclear fascin binds to Histone H3 when not associated with F-actin and suggests a dynamic exchange of fascin within the nucleus that may act to coordinate nuclear organisation.

## Nuclear fascin promotes efficient DDR

Assembly of nuclear F-actin occurs following replication stress and induction of DNA damage, and acts under these settings to regulate nuclear volume, chromatin organisation, and oxidation status, driving efficient DNA repair (*Belin et al., 2015*; *Lamm et al., 2020*). To determine whether nuclear fascin contributed to these functions, fascin KD HeLa cells expressing GFP or GFP-fascin were treated with neocarzinostatin (NCS), an ionising radiation mimetic, to directly induce DNA double-strand breaks (DSB), and levels of the key DDR factor γH2AX were then analysed by immunostaining (*Figure 3A*). Quantification of these data revealed a significant reduction in the immediate recruitment of γH2AX to damaged DNA foci in fascin-depleted cells (GFP) compared to GFP-fascin rescued cells at 30 and 60 min post-NCS treatment (*Figure 3B*). Interestingly, γH2AX levels significantly reduced in fascin expressing cells 120 min post-NCS addition upon resolution of DSB; however, fascin KD cells remained at a significantly higher level indicating sustained DDR in these cells (*Figure 3B*). When treated with the common chemotherapy drug cisplatin, which also induces DSB, fascin KD cells also exhibited reduced γH2AX levels compared to cells expressing WT fascin (*Figure 3C*). The reduced initial response to NCS in fascin-depleted cells was further verified in biochemically fractionated cells indicating enhanced γH2AX in nuclear fractions of NCS-treated GFP-fascin cells compared to fascin KD cells (*Figure 3D*; *Figure 3—source data 1*). Further analysis of GFP-fascin rescued cells demonstrated an increase in nuclear fascin and F-actin upon NCS treatment (*Figure 3E*). As DDR both depend upon and trigger changes in chromatin compaction status, we next analysed this using Histone H2B-H2B FRET as previously characterised by others (*Llères et al., 2009*) in control and fascin KD cells treated with NCS. Lifetime images (*Figure 3F*) and FRET efficiency calculations (*Figure 3G*) revealed a significant increase in FRET efficiency in fascin expressing cells upon NCS treatment indicating enhanced chromatin compaction (*Figure 3G*). However, fascin KD cells showed basally enhanced chromatin compaction with no significant change upon NCS treatment (*Figure 3G*). These data demonstrate that fascin is required for chromatin organisation and efficient DDR.

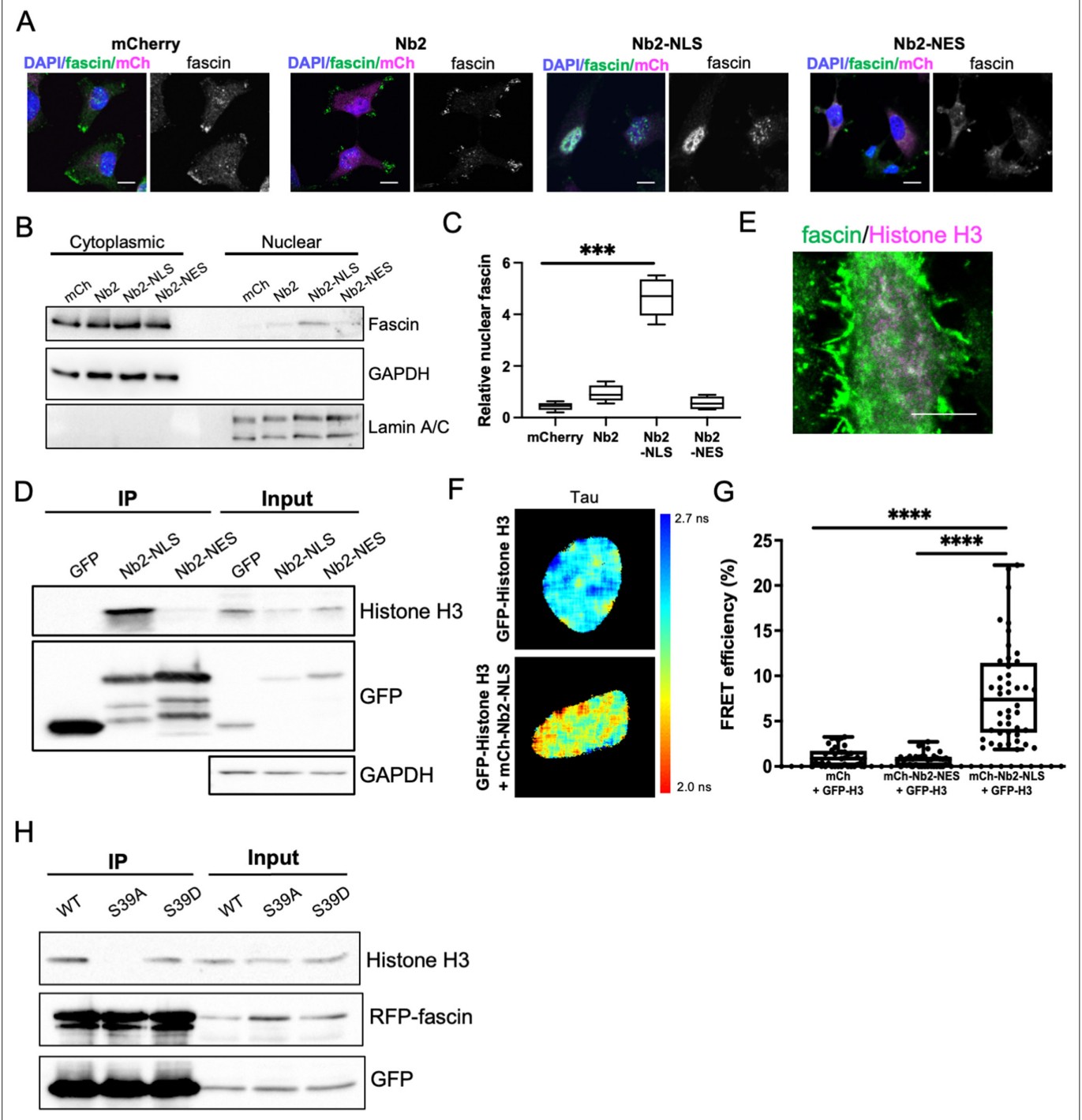

**Figure 2.** Nuclear fascin directly associates with Histone H3. (**A**) Representative confocal images of fascin knockdown (KD) MDA-MB-231 cells expressing GFP-fascin (green) and specified mCherry-Nb2 constructs (magenta) fixed and stained with DAPI (blue). Scale bars are 10 μm. (**B**) Representative western blot of fascin KD MDA-MB-231 cells expressing GFP or GFP-fascin (80 kDa) subjected to biochemical fractionation. Nuclear and cytoplasmic compartments probed for GAPDH (36 kDa) and Lamin A/C (69/62 kDa) . (**C**) Quantification of data from (**B**) from four independent experiments. (**D**) Representative western blot of HeLa cells expressing GFP or GFP-Nb2-nuclear localisation signal (NLS)/nuclear export signal (NES), subjected to GFP-Trap and probed for Histone H3 (15 kDa), GFP (25 kDa and ~50 kDa Nbs). Input shown on right; GAPDH as loading control (36 kDa). Representative of four independent experiments. (**E**) Representative confocal images of HeLa nuclei fixed and stained for fascin (green) and Histone H3 (magenta). Scale bar is 10 μm. (**F**) Representative images of HeLa cells expressing GFP-Histone H3 (top panels) or GFP-Histone H3 and mCherry-Nb2-NLS (bottom panels). Donor and acceptor channels shown; lifetime shown in far-right panels. Scale bars are 10 μm. (**G**) Quantification of fluorescence resonance energy transfer (FRET) efficiency from data as in (**F**) plus mCherry and mCherry-Nb2-NES acceptor controls. N=35 cells, from four independent

*Figure 2 continued on next page*

*Figure 2 continued*

experiments. Graph shows min/max and mean of dataset; each point represents a single cell. (**H**) Representative western blot of HeLa cells expressing RFP-fascin WT, S39A or S39D (80 kDa), and GFP-Nb2-NLS (~50 kDa), subjected to GFP-Trap and probed for specified proteins (Histone H3 is 15 kDa). Input shown on right. Representative of four independent experiments.

The online version of this article includes the following source data for figure 2:

**Source data 1.** *Figure 2B* full western blots.

**Source data 2.** Table detailing proteins identified by mass spectrometry associated with Nb2-NLS or Nb2-NES immunoprecipitated from HeLa cells.

**Source data 3.** *Figure 2D* full western blots.

**Source data 4.** *Figure 2H* full western blots.

## Sustained nuclear fascin reduces cell invasion and viability

To explore the functional consequences of enhanced and sustained nuclear fascin, we generated a doxycycline-inducible lentiviral version of the mCherry-tagged fascin-specific nanobodies coupled to NLS or NES sequences (as in *Figure 2*). This enabled precise triggering of expression of Nb2 in cells for sustained periods of time. We performed experiments in MDA-MB-231 cells as they display enhanced migratory and invasive capacity compared to HeLa and therefore represented a better model to assess these functions. Given that fascin is a canonical F-actin bundler known to promote filopodia formation, we evaluated filopodia assembly in live cells expressing Nb2 proteins for 48 hr. Filopodia were significantly reduced in Nb2-NLS expressing cells compared to controls, with no other changes seen in Nb2 or Nb2-NES expressing cells (*Figure 4A and B*). Similarly, cells expressing Nb2-NLS showed reduced migration speed in cells on 2D surfaces (*Figure 4C*) and reduced invasion into 3D collagen gels (*Figure 4D and E*; note higher nuclei numbers in 0 μm Z-section images of the Nb2-NLS samples indicating fewer cells able to invade) compared to control cells. Notably, Nb2-NLS expressing cells exhibited a similar reduction in invasion to fascin KD cells (*Figure 4E*), indicating that relocation of fascin to the nucleus prevents fascin function in the cytoplasm.

To determine longer-term consequences of forced nuclear fascin on cancer cell viability, we assessed proliferation over 96 hr post-Nb2 expression induction. Proliferation was significantly reduced in Nb2-NLS expressing cells compared to controls from 48 hr time periods onwards (*Figure 5A*) and this was coupled with a significant reduction in cell viability (*Figure 5B*). Further analysis of DNA content 48 hr post-Nb2 induction revealed that a significant proportion of cells expressing Nb2-NLS accumulated in the G0/G1 phase of the cell cycle relative to controls. Nb2-NLS expressing cells also exhibited reduced levels of S10 phosphorylated Histone H3 after 48 hr (*Figure 5D*) and enhanced cleaved caspase 3 by 72 hr post-induction (*Figure 5E*; *Figure 5—source data 1*). These findings collectively indicate that enhanced and sustained nuclear fascin specifically induces cell cycle arrest resulting in reduced viability and enhanced apoptosis.

## Unbiased high-content imaging reveals histone kinases as nuclear fascin regulators

Our data thus far demonstrate that fascin transport into the nucleus is dynamically regulated and contributes to nuclear F-actin bundling and DNA organisation and repair. However, as we also demonstrated that forced and sustained nuclear fascin can selectively drive apoptosis, we reasoned that identification of pathways that can control nuclear fascin may provide a foundation for therapies directed at tumour cell killing. To explore the pathways controlling nuclear fascin, we performed unbiased, imaging-based high-content phenotypic screening using fascin KD cells expressing mScarlet-fascin and the iRFP-nAC nuclear F-actin probe. Cells were treated with a library of ~13,000 annotated compounds, targeting multiple signalling pathways (*Parafati et al., 2020*), at a single concentration (5 μm) or neutral or positive (leptomycin B) controls for 24 hr, followed by fixation and imaging using high-content spinning disk confocal microscopy (*Figure 6A*). Resulting data were analysed using automated pipelines to define levels of nuclear fascin normalised to neutral and positive controls. The mean RZ' of the neutral and positive controls in this screen was 0.5017, indicating excellent assay robustness. This approach identified compounds with a range of activities including those that increased or decreased nuclear fascin relative to the neutral controls (*Figure 6B*, see *Figure 6—source data 1* for full dataset). We chose to focus on those that increased nuclear fascin and identified hits based on a minimum threshold value of 30% of the neutral controls, equivalent to ×3 the standard deviation

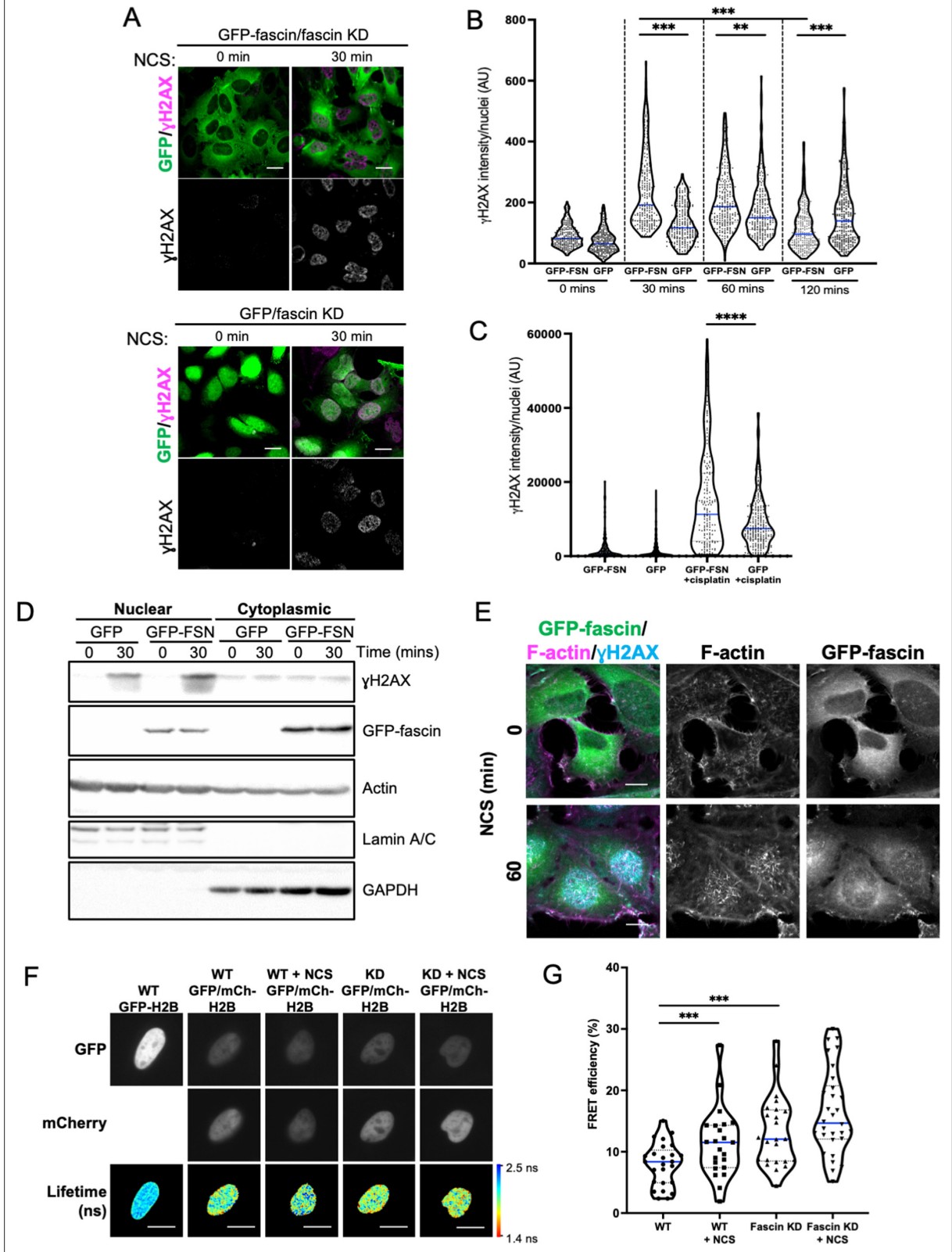

**Figure 3.** Nuclear fascin promotes efficient DNA damage response. (**A**) Representative confocal images of fascin knockdown (KD) HeLa cells expressing GFP or GFP-fascin (green) fixed and stained for γH2AX (magenta) before (0 min) or after (30 min) treatment with 0.5 μg/ml neocarzinostatin (NCS). Scale bars are 10 μm. (**B**) Quantification of γH2AX levels in cells from data as in (**A**). N=300–350 cells/condition, pooled from three independent experiments. (**C**) Quantification of γH2AX levels in cells treated with 5 μM cisplatin for 18 hr. N=240–300 cells/condition, pooled from three independent experiments.

*Figure 3 continued on next page*

*Figure 3 continued*

(**D**) Representative western blot of fascin KD HeLa cells expressing GFP (25 kDa) or GFP-fascin (FSN; 80 kDa) subjected to biochemical fractionation. Nuclear and cytoplasmic compartments probed for γH2AX (~15 kDa), Actin (42 kDa) and Lamic A/C (69/62 kDa). Representative of three independent experiments. (**E**) Representative confocal images of fascin KD HeLa cells expressing GFP-fascin (green) fixed and stained for γH2AX (cyan) and phalloidin (magenta) before (0 min) or after (60 min) treatment with NCS. Scale bars are 10 μm. (**F**) Representative images of WT or fascin KD HeLa cells expressing GFP-Histone H2B (left panels) or GFP-Histone H2B and mCherry-Histone H2B, with or without 30 min NCS treatment. Donor and acceptor channels shown; lifetime shown in bottom panels. Scale bars are 10 μm. (**G**) Quantification of fluorescence resonance energy transfer (FRET) efficiency from data as in (**F**). N=35 cells, pooled from three independent experiments. All graphs show min/max and mean of dataset; for figures in (**B**), (**C**), and (**G**), data is shown as violin plot with each point representing a single cell and mean shown in blue. **=p < 0.01, ***=p < 0.001, ****=p < 0.0001.

The online version of this article includes the following source data for figure 3:

**Source data 1.** *Figure 3D* full western blots.

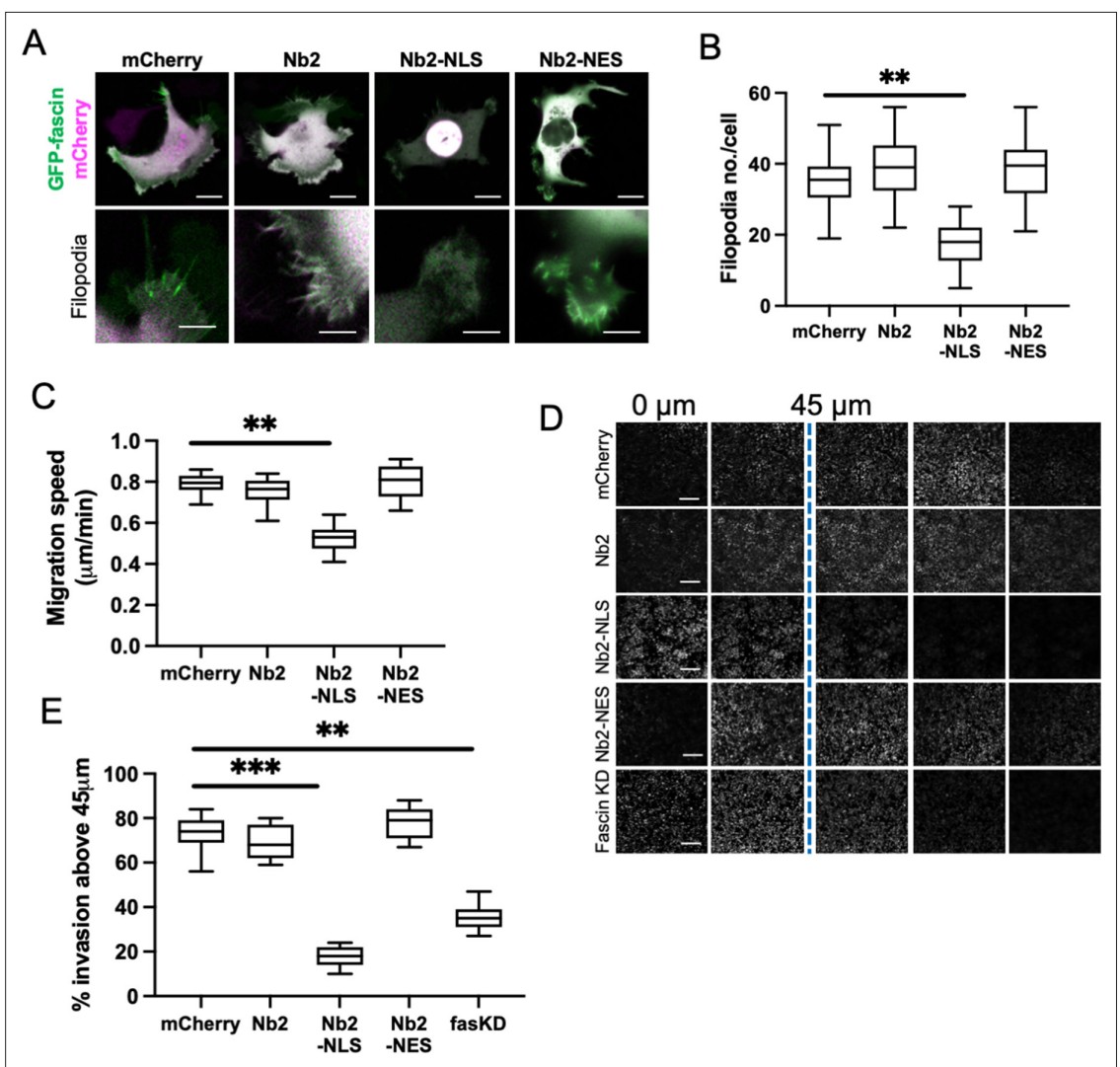

**Figure 4.** Sustained nuclear fascin reduces cell invasion.

(**A**) Representative confocal images of fascin knockdown (KD) MDA-MB-231 cells expressing GFP-fascin (green) and Nb2 constructs (magenta). Zoom region of filopodia shown below each. Scale bars are 10 μm. (**B**) Quantification of filopodia number/cell from data as in (**A**) from 30 cells, pooled from three independent experiments. (**C**) Quantification of 2D migration speed of cells as in (**A**) from 16 hr time-lapse movies. (**D**) Representative images of confocal Z-stacks from inverted invasion assays from cells as in (**A**), fixed after 48 hr invasion and stained for DAPI (shown). Fascin KD cells additionally shown (bottom row). (**E**) Quantification of invasion from data as in (**D**). Three wells imaged per experiment (3 fields of view/well), pooled from three independent experiments. All graphs show min/max and mean of dataset. **=p < 0.01, ***=p < 0.001, ****=p < 0.0001.

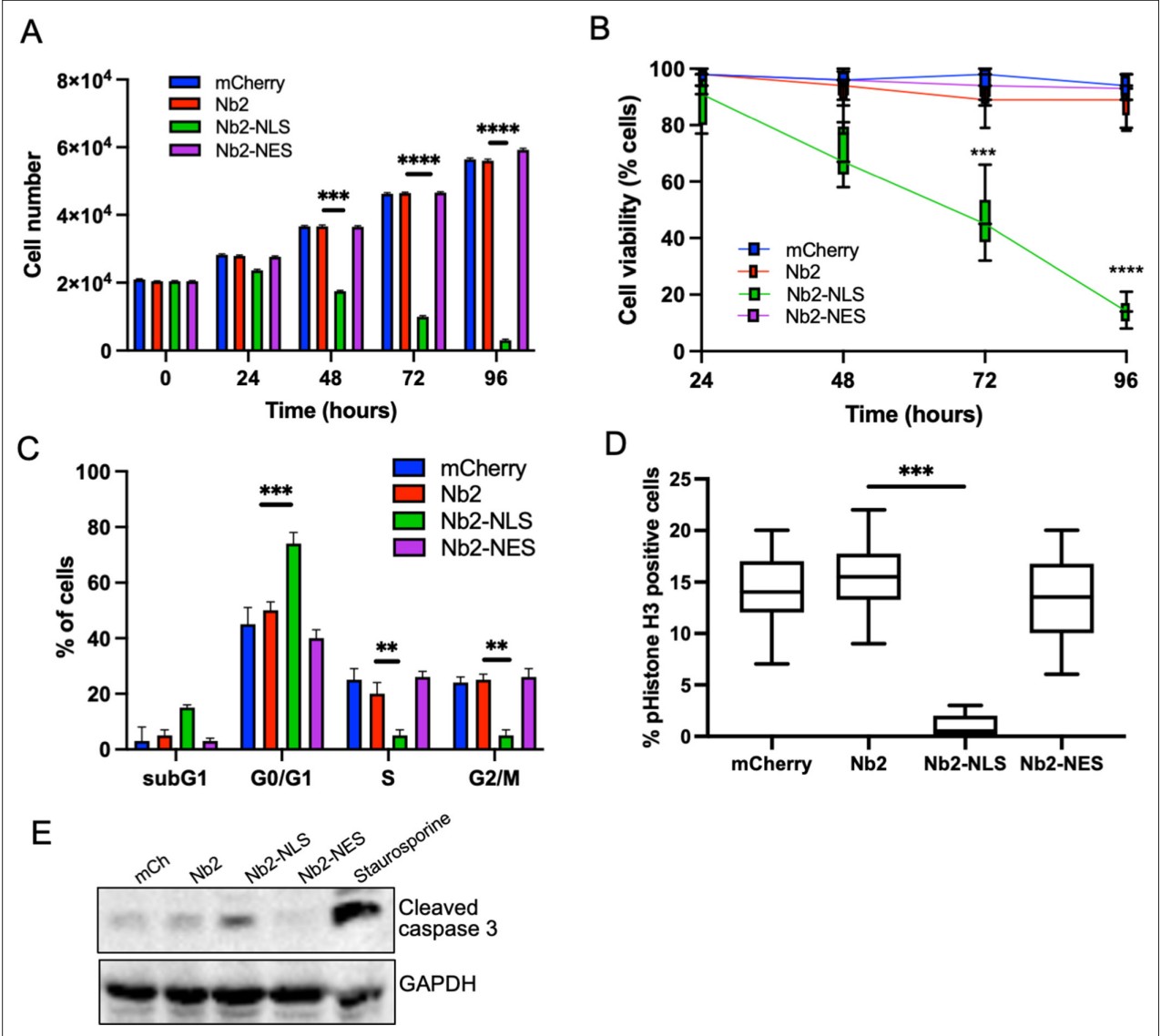

**Figure 5.** Forced and sustained nuclear fascin reduces cancer cell viability. (**A**) Quantification of proliferation of MDA-MB-231 cells expressing specified mCherry-Nb2 constructs over 96 hr. Cell nuclei counted from three replicate wells per condition per experiment. Mean ± SEM are shown. Representative of three independent experiments. (**B**) Quantification of cell viability using MTT assays of MDA-MB-231 cells expressing specified mCherry-Nb2 constructs over 96 hr. Three replicate wells per condition per experiment. Min/max and mean are shown. Representative of three independent experiments. (**C**) Quantification of cell cycle stage of MDA-MB-231 cells expressing specified mCherry-Nb2 constructs after 48 hr using FACS analysis of PI-stained cells. Three samples per condition were analysed. Mean ± SEM are shown. Representative of three independent experiments. (**D**) Quantification of pS10-Histone H3 staining from confocal images of MDA-MB-231 cells expressing specified mCherry-Nb2 constructs after 48 hr. N=90 cells analysed per condition, pooled from three independent experiments. Graph shows min/max and mean of dataset. ***=p < 0.001, ****=p < 0.0001. (**E**) Representative western blot of MDA-MB-231 cells expressing specified mCherry-Nb2 constructs after 72 hr probed for cleaved caspase 3 (~32 kDa) and GAPDH (36 kDa). Staurosporine (1 μm) treated cells (24 hr) were used as a positive control.

The online version of this article includes the following source data for figure 5:

**Source data 1.** *Figure 5E* full western blots.

of the neutral controls (*Figure 6B*; blue crosses above dashed line are hits). Following quality control checks (e.g., to exclude images with very few cells), 231 compounds were identified as hits. These compounds were taken forward for a repeat screen using the same cells to generate dose-response curves for each hit and control for any off-target or false positive findings. Data were analysed as for the primary screen (mean RZ'=0.5607) and nuclear fascin plotted as a function of compound concentration to identify AC50 values for each compound (*Figure 6C*; see *Figure 6—source data*

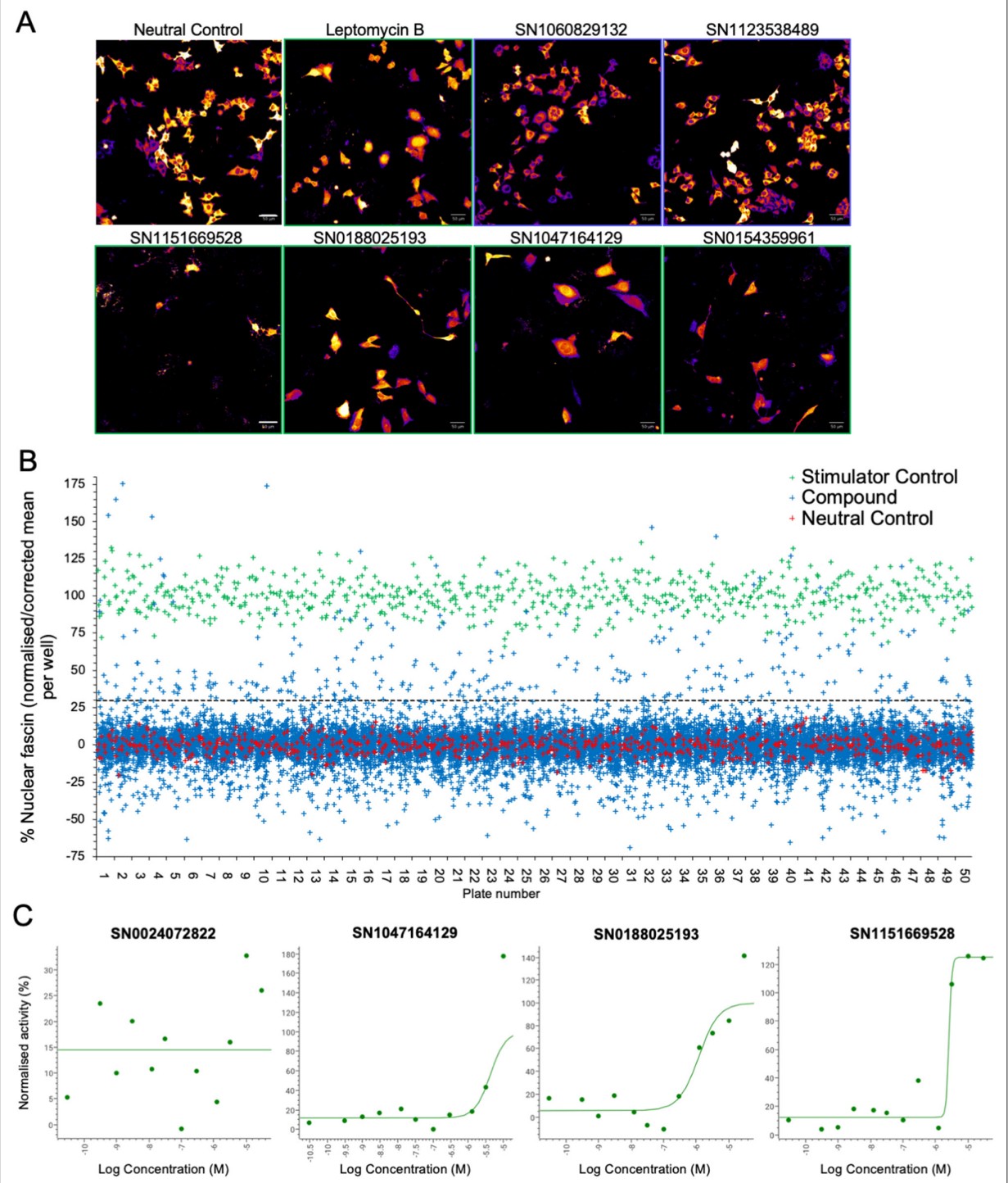

**Figure 6.** Identification of pathways controlling nuclear fascin.

(**A**) Example images from high-content screen showing mScarlet-fascin expressed in fascin knockdown (KD) cells (images shown in fire LUT for clarity) treated with specified compounds from small molecule library. Images framed in blue denote no increase in nuclear fascin; those in green denote high nuclear fascin. Scale bars are 50 µm. (**B**) Plot showing quantification of the mean % nuclear fascin/cell/well from each well of the entire screen (50×384 well plates). Data are normalised to neutral/stimulator controls and corrected for well position. Compound library data shown in blue, neutral controls in red and stimulator controls in green. (**C**) Representative compound response curves from selected compounds with normalised mean % nuclear fascin/cell/well plotted against compound concentration.

The online version of this article includes the following source data for figure 6:

**Source data 1.** Table detailing nuclear fascin and nuclear actin bundling levels in mScarlet-fascin/fascin knockdown (KD) HeLa cells treated with 12 increasing doses of the indicated compounds from the small molecule library.

*1* for full dataset). Nuclear F-actin was also analysed. This compound response analysis identified 96 compounds that were active against nuclear fascin, of which 43 also increased nuclear F-actin. The most highly represented pathways that were targeted by these compounds were cyclins and cyclin-dependent kinases (CDK). The promiscuous kinase glycogen synthase kinase 3 (GSK3) was also highly represented in the list of hits. Other target hits included inhibitors of IRAK4, haspin, TRK, and the proteasome.

To further refine the identified compounds and targets that enhanced nuclear fascin and F-actin in terms of functional importance, we performed an additional screen to detect apoptosis using a representative panel of 52 compounds that inhibited various pathways but all targeted fascin to the nucleus. Fascin KD HeLa cells expressing mScarlet or mScarlet-fascin were used for this analysis to identify fascin-dependent responses to compounds. Cells were treated with all compounds at a range of concentrations and imaged using automated high-content microscopy in the presence of a fluorescence-based caspase reporter (that fluoresced at 488 nm only upon onset of apoptosis) over 72 hr. Resulting data were quantified to produce apoptosis AC50 values for all compounds in both control and fascin KD cells (*Figure 7A*; see *Figure 7—source data 1* for full dataset). From these data, we identified 15 compounds that promoted apoptosis at lower concentrations in mScarlet-fascin expressing cells than in fascin KD cells, indicating dependence on enhanced and sustained nuclear fascin to drive apoptosis (*Figure 7B*). We chose to follow up on three hits that targeted CDK2 (SN0188025193), IRAK4 (SN1047164129), and haspin (SN1151669528, CHR-6494), as these compounds not only exhibited a greater differential apoptotic effect in fascin-expressing vs. fascin KD cells, but also induced nuclear fascin and F-actin in both the primary and compound response screens (*Figures 6A, C , and 7C* – magenta dots denote selected compounds; *Figure 7—source data 2*).

## Phosphorylation of Histone H3 controls nuclear fascin-H3 binding and DDR

Interestingly, all three identified targets have been shown to regulate histone phosphorylation and chromatin organisation (*Bhattacharjee et al., 2001*; *Dai et al., 2005*; *Liu et al., 2008*). We therefore hypothesised that these inhibitors may all be acting through a common pathway to prevent Histone H3 phosphorylation and thus disrupt fascin-Histone H3 association, promoting fascin nuclear retention and F-actin assembly. Treatment of cells with these three inhibitors revealed a total loss of pT3-Histone H3 phosphorylation in cells treated with haspin inhibitor for 4 hr and a reduction following treatment with IRAK4 inhibitor for 8 hr (*Figure 7D*; *Figure 7—source data 3*). CDK2 inhibition after 4 hr did not appear to directly alter Histone H3 phosphorylation (*Figure 7D*; *Figure 7—source data 3*). To determine whether these effects led to altered fascin-Histone H3 association, we performed GFP-Trap analysis of GFP-Nb2-NLS (as in *Figure 2D*) following treatment with each inhibitor and probed for Histone H3. Data revealed a striking reduction in fascin-Histone H3 association in cells treated with both IRAK4 and haspin inhibitors, but not with the CDK2 inhibitor (*Figure 7E*; *Figure 7—source data 4*). Analysis of immunoprecipitated GFP-fascin showed no change in fascin phosphorylation upon treatment with these inhibitors, suggesting these kinases do not act directly to post-translationally modify fascin (*Figure 7—source data 4*; *Figure 7—figure supplement 1*; *Figure 7—figure supplement 1—source data 1*). Further analysis revealed partial colocalisation between nuclear fascin and pT3-Histone H3 in untreated cells, coincident with assembly of nuclear F-actin (*Figure 7F*). To explore the relationship between fascin, nuclear F-actin, and Histone H3 upon haspin inhibition, cells co-expressing GFP-Histone H3, iRFP-nAC, and mScarlet-fascin were subjected to confocal time-lapse imaging in the presence or absence of haspin inhibitor. Haspin inhibition caused increased nuclear fascin, coupled with a strong induction of nuclear F-actin assembly that resulted in reorganisation of Histone H3 (*Figure 7G*). Treatment of control cells with haspin inhibitor also resulted in increased DDR as evidenced by γH2AX induction (*Figure 7H*), which correlated significantly with nuclear fascin levels (*Figure 7I*). These data collectively demonstrate that phosphorylation of Histone H3 promotes Histone H3-fascin binding and that a dynamic cycle of pT3-Histone H3 is required for fascin nuclear transport and efficient DDR.

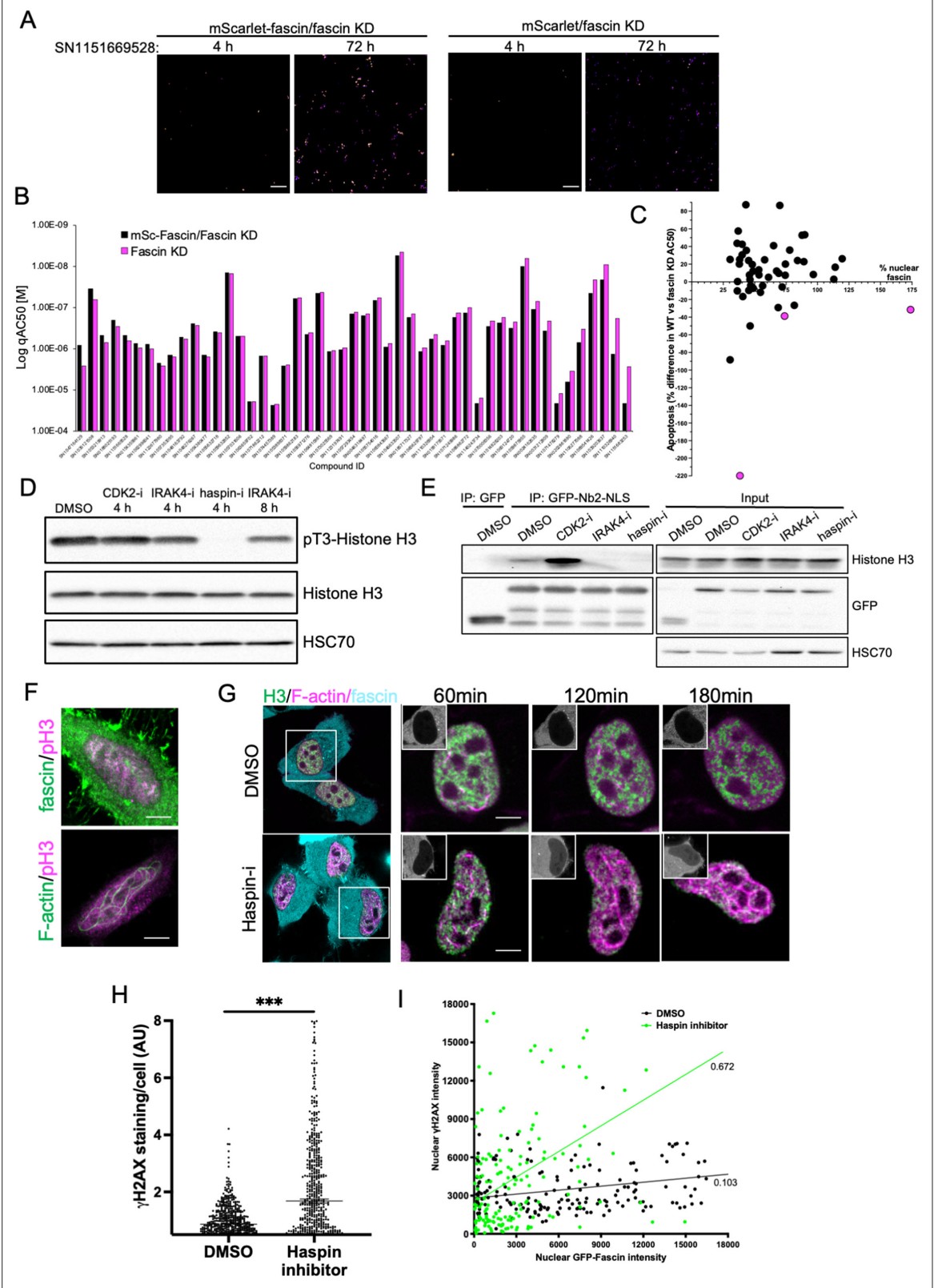

**Figure 7.** Haspin inhibition promotes sustained nuclear fascin and fascin-dependent apoptosis. (**A**) Example images from live cell time-lapse apoptosis screen in fascin knockdown (KD) HeLa cells expressing mScarlet-fascin or mScarlet, showing caspase reporter activity (images shown in fire LUT for clarity) after treatment with compound SN1151669528 (1.25 μM) for 4 or 72 hr. Scale bars are 150 μm. (**B**) Plot showing AC50 of specified compounds based on apoptosis screen data. (**C**) Correlation plot from apoptosis screen data showing % difference in apoptosis AC50 between fascin KD HeLa

*Figure 7 continued on next page*

*Figure 7 continued*

cells expressing mScarlet-fascin (WT) or mScarlet (KD), plotted as a function of mean % nuclear fascin/cell/well from original screen (as in *Figure 6B*), after treatment with hit compounds. Magenta data points denote hits selected with significantly lower death in KD cells vs. WT. (**D**) Representative western blot of HeLa cells treated with target inhibitors (1.25 µM) probed for pT3/total Histone H3 (15 kDa) and HSC70 (70 kDa). Representative of three independent experiments. (**E**) Representative western blots of HeLa cells expressing GFP or GFP-Nb2-NLS treated with specified compounds (1.25 µM, 4 hr) followed by GFP-Trap and probing for Histone H3 (15 kDa), GFP (80 kDa for GFP-FSN and 25 kDa for GFP) and HSC70 (70 kDa). Representative of three independent experiments. (**F**) Example confocal images of HeLa cells expressing iRFP-nAC nuclear actin probe, stained for fascin and pT3-Histone H3 (magenta). Scale bars are 10 µm. (**G**) Representative images of HeLa cells co-expressing GFP-Histone H3 (green), iRFP-nAC (magenta), and mScarlet-fascin (cyan) treated with DMSO or haspin inhibitor followed by time-lapse imaging. First panels show whole cells at time 0, subsequent panels show single nuclei with mScarlet-fascin as inset panels at specified time points. Scale bars are 10 µm. (**H**) γH2AX levels in cells treated with DMSO or 1.25 µM haspin inhibitor for 4 hr. Each point represents a single cell; representative of three independent experiments. ***=p < 0.001. (**I**) Correlation plot between nuclear GFP-fascin and γH2AX intensity from data as in (**H**). Values represent linear regression slope. NLS, nuclear localisation signal.

The online version of this article includes the following source data and figure supplement(s) for figure 7:

**Source data 1.** Table detailing apoptosis levels in mScarlet-fascin/fascin knockdown (KD) (WT) or mScarlet/fascin KD (KD) HeLa cells treated with 12 increasing doses of the indicated compounds from the small molecule library for 72 hr.

**Source data 2.** Table detailing nuclear fascin and nuclear actin bundling levels in mScarlet-fascin/fascin knockdown (KD) HeLa cells treated with 12 increasing doses of the indicated compounds from the small molecule library.

**Source data 3.** *Figure 7D* full western blots.

**Source data 4.** *Figure 7E* full western blots.

**Figure supplement 1.** Haspin and IRAK4 do not promote fascin phosphorylation.

**Figure supplement 1—source data 1.** Figure Supplement 2 full western blots.

# Discussion

Fascin has been widely studied as an important regulator of F-actin bundling leading to enhanced filopodia assembly and invasion in cancer cells. Our study provides insight into a new role for fascin in bundling F-actin within the nucleus to support chromatin organisation and efficient DDR, representing additional important routes through which fascin can support tumourigenesis. We reveal that fascin associates directly with phosphorylated Histone H3 leading to regulated levels of nuclear fascin to support these phenotypes. Forcing nuclear fascin accumulation through the expression of nuclear-targeted fascin-specific nanobodies or inhibition of kinases that promote Histone H3 phosphorylation results in enhanced and sustained F-actin bundling leading to reduced invasion, viability, and nuclear fascin-specific/driven apoptosis.

We found that fascin bundles F-actin in the nucleus. Fascin therefore joins several other actin-associated proteins, including cofilin and profilin, that have been shown to localise to the nucleus and regulate nuclear F-actin assembly (*Caridi et al., 2019*). Cofilin is actively imported into the nucleus in complex with actin monomers by importin 9 (*Dopie et al., 2012*), whereas profilin binds to actin during nuclear export by exportin 6 (*Stüven et al., 2003*). The exact mechanism through which fascin is imported and exported from the nucleus remains unclear, but we have established that fascin contains an NLS and NES and is blocked from nuclear export by leptomycin B, an inhibitor of exportin 1. Interestingly, our screen for nuclear fascin regulators identified two other exportin 1 inhibitors that also promoted nuclear fascin, selinexor and KPT-276, providing further evidence that exportin 1 may be involved in export of nuclear fascin. However, further studies would be required to verify whether this is a direct or indirect effect. We have previously identified that fascin localises to the NE and binds directly to the NE protein nesprin-2 (*Jayo et al., 2016*). It would be of interest to determine whether this interaction serves as a pool of fascin for nuclear import in response to DDR or other external stress, including mechanical triggers. Another NE-associated protein, ATR, has a well-established role in DDR and regulates chromatin compaction in response to mechanical stress (*Kumar et al., 2014*). Interestingly, ATR has also been recently shown to associate with nesprin at the NE (*Kidiyoor et al., 2020*). Examining the potential co-operation between fascin and ATR in mediating DDR would be an interesting avenue to pursue in future studies.

Regardless of the mechanism through which fascin is imported and exported to the nucleus, temporal regulation is vital to the homeostatic nuclear function of fascin. Following cytokinesis, fascin transiently localises to the nucleus, where it bundles F-actin. Nuclear F-actin has previously been shown to be involved in chromatin decondensation during nuclear expansion in G1, in a process that

is terminated by the actin depolymerising protein cofilin (*Baarlink et al., 2017*). Given that basal levels of chromatin compaction were elevated in fascin-depleted cells (*Figure 3G*), our data indicate that fascin is responsible for initiating F-actin-dependent chromatin decompaction. Following nuclear F-actin assembly, fascin is exported from the nucleus, and the importance of this shuttling is highlighted by our finding that forced and sustained nuclear fascin leads to apoptosis.

Histone H3 appears to play an important role in the spatiotemporal regulation of nuclear fascin. Fascin has previously been found to colocalise with Histone H3 Lys4 trimethylation by ChIP-seq (*Saad et al., 2016*) and our data indicate that fascin directly interacts with Histone H3. The association of fascin with Histone H3 appears to be dependent on Histone H3 phosphorylation. It is possible that phosphorylation may alter the conformation of Histone H3 to allow fascin to bind. The binding site of fascin to Histone H3 is unknown, but if Histone H3 binding acts as a scaffold for F-actin bundling by fascin, this would suggest that the binding site on fascin is distinct from the two F-actin bundling sites (S39 and S274). However, we found that inhibition of Histone H3 phosphorylation led to enhanced fascin nuclear accumulation and F-actin assembly, indicating that the bundling of nuclear F-actin by fascin is not necessarily dependent on phospho-Histone H3. It is possible that fascin-dependent F-actin bundling requires an initial scaffold on Histone H3 but becomes disrupted and assembles elsewhere in the nucleus under non-homeostatic conditions. One possibility is that F-actin binding could block access to the C-terminal NES of fascin resulting in the nuclear retention of fascin and perpetuation of F-actin bundles.

In addition to regulating chromatin decondensation post-mitosis, nuclear F-actin has also shown by others to reorganise chromatin for effective DDR. We observed that DDR was impaired in fascin KD cells, indicating that the function of F-actin in DDR is dependent on bundling by fascin. Presumably, F-actin must disassemble and fascin be exported from the nucleus upon successful resolution of DNA damage. Hence, if fascin is forced into the nucleus, the resultant perpetuation of nuclear F-actin may prevent DDR from being efficiently resolved. This would likely lead to growth inhibition and cell cycle defects and could explain why nanobodies or inhibitors that target fascin to the nucleus cause cell death.

Our high-content imaging screen identified pathways that are involved in regulating nuclear fascin accumulation. Haspin inhibition was particularly effective in driving nuclear accumulation of fascin and F-actin and driving fascin-dependent apoptosis. Haspin mediates Histone H3 phosphorylation prior to mitosis in response to activation by cyclin B/CDK1 and has been implicated in tumourigenesis. Haspin is required for chromosomal stability; hence, treatment with the haspin inhibitor CHR-6494 has been identified to induce apoptosis in cancer cells by inducing genomic instability (*Quadri et al., 2022*). Our data suggest that prolonged nuclear accumulation of fascin in response to haspin inhibition may also contribute to these phenotypes.

Other pathways that we identified to regulate nuclear fascin and F-actin in our high-content imaging screen include cyclins, CDKs, and GSK3. Since we have established that fascin and F-actin transiently accumulate in the nucleus post-mitosis, it is logical to expect that their activities are dynamically controlled by cell cycle regulators such as cyclins and CDKs. However, it is also likely that any inhibitors that induce cell cycle arrest in G1 would also be identified as hits non-specifically, as fascin and F-actin are elevated at this cell cycle stage by default. Further experiments will therefore be required to verify whether these compounds affect fascin directly or indirectly and to establish their exact mechanisms of action.

The pathways that contribute to nuclear F-actin bundling have previously been shown to involve factors such as GPCRs and calcium signalling (*Wang et al., 2019*). In addition to the fascin-dependent F-actin bundling pathways that we have described, we also identified multiple other inhibitors that either promoted or inhibited nuclear F-actin bundling. Similarly, although we focussed on targets that promoted nuclear fascin in this study, our screen also identified multiple compounds that inhibited nuclear fascin accumulation. Hence, further studies investigating the mechanism of action of these compounds could shed further light on the pathways that regulate both nuclear fascin and F-actin in the nucleus.

In summary, our study provides insight into a new role for fascin in controlling nuclear F-actin bundling to support tumour cell viability. Given the broad overexpression of fascin reported in solid tumours, this molecule is an interesting therapeutic target but has yet to be convincingly blocked directly using small molecules or other modalities. We propose that promoting fascin nuclear

accumulation using targeted inhibitors such as those against haspin or IRAK4 may provide means to drive nuclear F-actin accumulation in these cells and initiate apoptosis. This would provide an alternative, albeit less direct, approach to target fascin function both in the nucleus and cytoplasm and suppress invasion while simultaneously driving nuclear F-actin accumulation, and thereby prevent tumour growth and metastasis.

## Materials and methods

### Antibodies and reagents

The following antibodies were used: fascin (55K2; MAB3582), FLAG (M2; F3165), GFP (11814460001), and pT3 Histone H3 (07-424) were from Merck; Actin (ab6276) and Lamin A/C (ab8984) were from Abcam; Histone H3 (2128A; MAB9448) and γH2AX (NB100-384) were from Novus; cleaved Caspase-3 (5A1E; 9664) and pS10 Histone H3 (9701) were from Cell Signaling Technology; GAPDH (6C5; GTX28245) was from GeneTex; and HSC70 (sc-7298) was from Santa Cruz. The following reagents were used: leptomycin B was from Cambridge Bioscience, NCS was from Merck, cisplatin was from Tocris/Biotechne, and Pro-Q Diamond was from Thermo Fisher.

### DNA constructs

The following plasmids were used: GFP and mCherry (Clontech/Takara Bio); GFP- and RFP-fascin WT/S39A/S39D (*Zanet et al., 2012*), NLS (PKKKRKV), or NES (NLVDLQKKLEELELDEQQ) coding sequences were inserted C-terminal to GFP-fascin WT to create GFP-fascin NLS and NES; mScarlet and mScarlet-fascin (*Pfisterer et al., 2020*) were subcloned into pLNT/Sffv; Histone H2B-GFP and mCherry-H2B (*Llères et al., 2009*) were a gift from Angus Lamond (University of Dundee); GFP-Histone H3 (*Kimura and Cook, 2001*) was a gift from Hiroshi Kimura (Tokyo Tech). GFP-Nb2/NLS/NES (*Van Audenhove et al., 2014*) were subcloned into the pmCherry-N1 vector, which was then subcloned into pLVX for doxycycline-inducible expression. Fascin Nb2 with an NLS or NES was generated in the following manner. A pLVX tight puro vector that drives expression of the gelsolin nb11 nanobody (*Van Audenhove et al., 2013*) containing a C-terminal SV40 large T-antigen NLS (PKKKRKV) or C-terminal MAPKK NES (NLVDLQKKLEELELDEQQ) was cut with BamHI and EcoRI to excise the gelsolin nb. Fascin Nb2 was amplified using primers FWD 5' AGGGATCCACCGGTCGCCACCATG CAGGTGCAGCTGCAGGAG 3' and REV 5' TGGTGGCGACCGGTGGATCCCTGC 3' with the pHEN4 plasmid containing Fascin Nb2 as template. The cDNA was subsequently cloned into the pLVX tight puro vector using the Cold Fusion cloning kit (System Biosciences), according to the manufacturer's instructions. Similarly, to obtain NES or NLS-Fascin Nb2 in the pmCherry vector (Clontech), gelsolin nb11 (*Van Audenhove et al., 2013*) was excised from the vector (already containing the targeting sequences) with BamHI and EcoRI restriction enzymes and replaced by Fascin Nb2, PCR amplified from the pHEN4 vector with primers FWD 5' AGCTCAAGCTTCGAATTCATGCAGGTGCAGCTGCAG GAG 3' and REV 5' GGCGACCGGTGGATCCTTACTTGTACAGCTGCTGCT 3'. iRFP-nAC was created by subcloning iRFP from piRFP (a gift from Vladislav Verkhusha, Addgene plasmid # 31857) into GFP-nAC (ChromoTek) using a Q5 site-directed mutagenesis kit (New England Biolabs), according to the manufacturer's instructions. A BamH1 site was first added to GFP-nAC using FWD 5' AGGATCCA TGAGCGGGGGCGAGGAG 3' and REV 5' TCAGCCATAGAACCTCCTCCACCGCTAC 3' primers. A KpnI site was then added using FWD 5' GGTACCGGGCCGCCTAAGAAAAAG 3' and REV 5' CCTG TACAGCTCGTCCAT 3' primers. The resultant vector was digested BamH1-KpnI to remove GFP, which was then replaced with iRFP from piRFP digested with BamH1-KpnI. The stop codon and KpnI site on iRFP were deleted using FWD 5' GGGCCGCCTAAGAAAAAG 3' and REV 5' CTCTTCCATCACGCCG AT 3' primers. FLAG-NLS-Actin was a gift from Bernd Knoll (*Kokai et al., 2014*).

### Cell lines and transfection

Human HeLa, MDA-MB-231, and HEK293 cells were obtained from ATCC and cultured at 37°C, 5% $CO_2$ in high-glucose DMEM (Merck) supplemented with 10% FBS (Thermo Fisher), 1% glutamine, and 1% penicillin/streptomycin (Merck). Fascin KD cells were selected and maintained using puromycin, and cells were sub-cultured using 0.05% trypsin-EDTA (Merck) in PBS. For expression of fascin (transient or stable), we used fascin KD HeLa cells and re-expressed fascin at physiological levels. Transient expression was achieved using Lipofectamine 2000 (Thermo Fisher). Stable fascin HeLa and

MDA-MB-231 cell line generation has been described previously (*Jayo et al., 2016*; *Pfisterer et al., 2020*). All cells were tested monthly for mycoplasma.

## GFP-Trap immunoprecipitation

Cells expressing GFP-tagged proteins were washed with ice-cold PBS then lysed in cold lysis buffer (50 mM Tris pH 7.4, 150 mM NaCl, 0.5% TX-100 plus protease and phosphatase inhibitors). Lysates were then centrifuged at 16,000× *g* for 10 min at 4°C. GFP-Trap beads (ChromoTek) were washed ×3 in lysis buffer then incubated with cell lysate supernatants at 4°C for 2 hr with rotation. Thirty μl of supernatants were also set aside as input. Beads were then washed ×3 with lysis buffer; ×2 sample buffer was added to the beads and boiled for 5 min at 95°C prior to western blot analysis.

## Cell fractionation

Cells were washed in PBS then lysed in hypotonic buffer (20 mM Tris pH 7.5, 10 mM NaCl, 3 mM MgCl$_2$, 0.5% NP-40 plus). Lysates were centrifuged at 3000× *g* for 5 min to pellet nuclei. Supernatant was removed and centrifuged at 20,000× *g* for 15 min and the resultant supernatant contained the cytosolic fraction. The nuclear pellet was washed ×3 in hypotonic buffer then resuspended in 50 mM Tris pH 7.5, 100 mM NaCl, 5 mM MgCl$_2$, 5 mM CaCl$_2$, 2% SDS plus protease and phosphatase inhibitors, and centrifuged at 20,000× *g* for 15 min. The resultant supernatant contained the nuclear fraction. Cytosolic and nuclear fractions were diluted in sample buffer and analysed by western blot.

## Western blotting

Western blotting was performed as described previously (*Jayo et al., 2016*). Briefly, cells were lysed in sample buffer and proteins were separated under reducing conditions by SDS-PAGE, blotted onto PVDF membranes, blocked with 5% BSA or milk and probed for specified proteins. Proteins were then detected with ECL chemiluminescence kit (Bio-Rad Laboratories) and imaged (ChemiDoc Imaging Systems, Bio-Rad Laboratories). Blots were analysed and processed using Image Lab (v6.0.1, Bio-Rad Laboratories).

## Proteomics

Cells expressing GFP-tagged proteins were lysed and subjected to GFP-Trap immunoprecipitation (IP), then separated by SDS-PAGE. Gels were then stained using a Pierce Silver Stain Kit (Thermo Fisher), according to the manufacturer's instructions. Bands of interest that were differentially present between NLS and NES Nb samples were excised from silver-stained gels then subjected to in-gel reduction, alkylation, and digestion with trypsin prior to analysis by mass spectrometry. Cysteine residues were reduced with dithiothreitol and derivatised by treatment with iodoacetamide to form stable carbamidomethyl derivatives. Trypsin digestion was carried out overnight at room temperature after initial incubation at 37°C for 2 hr. Peptides were extracted from the gel pieces by a series of acetonitrile and aqueous washes. The extract was pooled with the initial supernatant and lyophilised. The sample was then resuspended in 10 μl of resuspension buffer (2% acetonitrile in 0.05% formic acid) and analysed by LC-MS/MS. Chromatographic separation was performed using a U3000 UHPLC NanoLC system (Thermo Fisher). Peptides were resolved by reversed phase chromatography on a 75 μm C18 column (50 cm length) using a three-step linear gradient of 80% acetonitrile in 0.1% formic acid. The gradient was delivered to elute the peptides at a flow rate of 250 nl/min over 60 min. The eluate was ionised by electrospray ionisation using an Orbitrap Fusion Lumos (Thermo Fisher) operating under Xcalibur v4.1.5. The instrument was programmed to acquire in automated data-dependent switching mode, selecting precursor ions based on their intensity for sequencing by collision-induced fragmentation using a TopN CID method. The MS/MS analyses were conducted using collision energy profiles that were chosen based on the mass-to-charge ratio (m/z) and the charge state of the peptide. Raw mass spectrometry data were processed into peak list files using Proteome Discoverer (v2.2, Thermo Fisher). The raw data files were processed and searched using the Mascot (v2.6.0, Matrix Science) and Sequest (University of Washington) search algorithms against the Human Taxonomy database (Uniprot). Full proteomics reporting details are provided in *Figure 2—source data 2*. Resulting data and subsequent follow-up validation experiments were prioritised based on number of unique peptides identified and consistent identification in NLS rather than NES Nb2 IP samples.

## Cell synchronisation

Cells were incubated overnight then treated with 2 mM thymidine (Merck) for 24 hr. Cells were then washed and incubated in fresh media for a further 10 hr prior to fixation.

## Immunofluorescence

Cells were fixed in either cold methanol for 2 min or 4% paraformaldehyde (PFA) for 15 min at room temperature, then washed ×3 in PBS or TBS and permeabilised in 0.1% TX-100 in PBS or TBS for 10 min at room temperature. Cells were washed ×3 in PBS or TBS followed by blocking with 3% BSA in PBS or TBS and incubation with specific primary antibodies in blocking solution for 1 hr. Cells were then washed ×3 in PBS or TBS and incubated with fluorescent secondary antibodies and dyes (DAPI, Cell Signaling Technology; Phalloidin, Thermo Fisher) in blocking solution. After washing, samples were mounted with Fluorsave (Merck).

## Microscopy

Confocal microscopy images were acquired on a Nikon A1R inverted confocal microscope (Nikon Instruments UK) with an environmental chamber maintained at 37°C. Images were taken using a ×40 or ×60 Plan Fluor oil immersion objective (numerical aperture of 1 and 1.4, respectively). Excitation wavelengths of 405, 488, 561, or 640 nm (all diode laser) were used. In experiments where multiple cell lines/conditions were analysed from the same cell type, all images were acquired at identical laser settings to permit comparison of intensities. Images were acquired using NIS-Elements imaging software (v4) and were saved in Nikon Elements in the .ND2 format. Further image processing was carried out in Fiji processing software (*Schindelin et al., 2012*).

## Nuclear actin quantification

Images of iRFP-nAC expressing cells were captured by confocal microscopy and analysed for nuclear F-actin structure using a custom-designed MATLAB script. Nuclear actin structures were analysed using a custom MATLAB script with the aim quantifying the degree to which nuclear actin was organised in filaments as opposed to more diffuse aggregate structures. Individual nuclei were selected from the fluorescence image by a rectangular ROI. For each cropped ROI the background level was estimated by histogramming the pixel intensities and finding the turning point. Pixel intensities below this threshold were judged to be background and set to zero. The foreground structure was then identified by a process of constrained grey-scale erosion. A 3×3 kernel was rastered through the image identifying the lowest intensity non-zero pixel which was then removed (set to zero) provided it would not result in fewer than 3 remaining non-zero pixels in a constrained positional relationship in the 3×3 region. The process was repeated several times until no further removal of foreground pixels was possible without breaking this condition. The positional constraint enabled the removal of one more pixel from the foreground in the 3×3 patch if the three formed an angle of less than 90° including the centre pixel, this biases the detection of foreground to more linear structures such as fibres over sparse clusters. The resultant image was then binarised and subjected to one round of dilation and erosion (morphological closing) representing the extracted foreground structure. The degree to which the foreground consisted of linear fibres was parameterised by measuring the proportion of foreground pixels removed by a subsequent skeletonisation operation. The more the foreground was made of linear structures, the lower the proportion removed by this operation. Images were processed in MATLAB R2020a (Mathworks) and the quantitative analysis was performed in MATLAB and GraphPad Prism (GraphPad). MATLAB script is available at https://github.com/IJAL98/NucActin.

## FRET/FLIM and data analysis

Imaging was performed on a confocal microscope (Ti Eclipse, Nikon) with a ×40 objective throughout (Plan Fluor NA 1.3; DIC H, WD 0.2; Nikon), with fluorescence lifetime acquired by time-correlated single-photon counting electronics (SPC-830) on a DCC-100 control (both Becker & Hickl). Acquisition was performed at 920 nm laser excitation (MaiTai, DeepSee; Spectra-Physics) for 5 min to collect sufficient photons for fitting, while avoiding pulse pile up or significant photobleaching. Corresponding widefield images were taken for GFP and mCherry (DS-Qi1Mc camera; Nikon). Lifetime raw data were analysed with TRI2 (*Barber et al., 2009*) and mono-exponential fitting was used. Histogram data are plotted as mean FRET efficiency (calculated based on GFP lifetime without acceptor) from the

indicated number of cells per sample. FLIM images are presented in reverse rainbow pseudocolour with red representing low lifetime and high FRET/interaction and blue representing high lifetime and low FRET/interaction.

## Cell invasion

Inverted invasion assays into Collagen I gels were performed as previously described (*Jayo et al., 2016*). Briefly, 1.6 mg/ml collagen gel (Corning) supplemented with 10 µg/ml fibronectin (Merck) and 2% FBS (Thermo Fisher) was prepared and 150 µl was poured inside each transwell insert with 8 µm wide pores (Greiner Bio-One) and left to polymerise for 2 hr at room temperature. Transwells were inverted and 50 µl of MDA-MB-231 cells were plated on top of each at $2-20 \times 10^5$ cells/ml. Cells were left to adhere for 3 hr and transwells re-inverted. Serum-free Ham F-12 media (Merck) was added to the lower well, and 20 ml of 10% FBS containing Ham F-12 media was added inside the insert. Cells were left to invade for 72 hr and nuclei were stained for 5 min prior to imaging. Transwell inserts were placed in a 35 mm live cell imaging dish and confocal sections were taken every 1.5 µm in five independent fields per transwell with a ×20 dry objective in a Nikon Eclipse Ti-E inverted microscope with an A1R Si Confocal system. To quantify invasion levels a Fiji software-based macro was used to perform all the analysis in batch mode. Briefly, a Gaussian filter was applied to all the channels (sigma value of 2) and the nuclear channel was thresholded, binarised, and watershed applied. The number of nuclei was quantified after reducing the slice number by a factor of 6. To establish the starting position among different stacks, the first image with more than five intact nuclei was considered as z=0 and all subsequent invasive depth measurements calculated from that point.

## Cell proliferation and viability analysis

Equal numbers of cells were plated on a 24-well plate and incubated for specified times. Cells were then washed with PBS, before fixing with 4% PFA/PBS for 10 min. Nuclei were stained with DAPI to enable quantification. Cells were imaged on an EVOS FL Auto 2 fluorescent microscope (Thermo Fisher). 9×9 tile scans were obtained using a ×10 air objective with 3.2 MP CMOS camera. Excitation using DAPI LED light cube was used. Images were acquired using EVOS software (v2). Tiles were knitted into TIFF files using Fiji software and total cell count was obtained by thresholding for nuclear stain followed by automated counting. Using three biological repeats per condition, an average number of cells per condition was calculated. Viability assays were conducted in 96-well plates using MTT assays according to the manufacturer's instructions (Merck; TOX1-1KT).

## Cell cycle analysis

Cells were fixed in ice-cold 70% ethanol for 30 min followed by treatment with ribonuclease (100 µg/ml stock) at 4°C, washing and incubation with propidium iodide (50 µg/ml stock solution; Merck) for 30 min at room temperature. Detection was performed using an Attune NxT flow cytometer (Thermo Fisher) with 488 nm excitation, and the resulting data was analysed using ModFit LT to detect DNA content Gaussian distributions and determine % of cells within each cell cycle phase.

## Quantification of filopodia

Filopodia were quantified using the MATLAB plugin CellGeo (*Tsygankov et al., 2014*). Briefly, live cell confocal images were thresholded using MovThresh, followed by BisectoGraph, which maps arbitrarily a polygon on a tree graph required for FiloTrack, which identifies filopodia length and number.

## Phenotypic high-content screen and analysis

The compound library used in the screen consisted of ~13,000 target-annotated compounds in DMSO, as previously described (*Parafati et al., 2020*). mScarlet-fascin/fascin KD HeLa cells were transfected with iRFP-nAC, then plated in CellCarrier Ultra 384 well plates (PerkinElmer) at a concentration of 4400 cells/well. After 24 hr, 10 nM leptomycin b, vehicle control, or library compounds (*Figure 6—source data 1*) were added using an Echo liquid handler (Labcyte/Beckman Coulter). Cells were incubated for a further 24 hr then fixed in 4% PFA and stained with DAPI and Alexa Fluor 488 Phalloidin (Thermo Fisher). Plates were imaged using a ×20 objective on a CellVoyager CV8000 High-Content Screening System (Yokogawa). Images were analysed using Columbus (PerkinElmer)

and Genedata Screener (Genedata) software. All high-content imaging data is deposited to the Image Data Resource (https://idr.openmicroscopy.org) under accession number idr0139.

Apoptosis assay and analysis mScarlet-fascin/fascin KD or mScarlet/fascin KD HeLa cells were plated with 2 mM CellEvent Caspase-3/7 Green Detection Reagent (Thermo Fisher) in CellCarrier Ultra 384 well plates (PerkinElmer) at a concentration of 1000 cells/well. After 24 hr, 1 µM staurosporine (Merck), vehicle control, or compounds from the small molecule library (*Figure 7—source data 1*) were added using an Echo liquid handler (Labcyte/Beckman Coulter). Cells were then incubated in an IncuCyte Zoom (Sartorius) at 37°C and imaged at ×10 every 4 hr for 72 hr. Data were analysed using IncuCyte Zoom (Sartorius) and Genedata Screener (Genedata) software. All raw apoptosis image data is deposited to the Image Data Resource (https://idr.openmicroscopy.org) under accession number idr0139.

## Statistical analysis

All experiments were repeated at least three times unless indicated otherwise. All statistical tests were performed using GraphPad Prism software (GraphPad). Outliers were removed following the ROUT method for outlier removal from non-linear regressions. When the populations followed a parametric distribution, ANOVA followed by Dunnet's multiple comparison test or Student's t test for two groups' mean comparison was used. Otherwise, data not following normal distributions were analysed with non-parametric Kruskal-Wallis test followed by Dunn's multiple comparison test. Correlation analysis (*Figure 7I*) was performed using Pearson's correlation coefficient analysis in GraphPad Prism (used to interpret the confidence interval of r and the p value testing the null hypothesis that there is no correlation between the two variables and any correlation observed is a consequence of random sampling). Data are expressed as means ± SEM. Significance was taken as $p < 0.05$ (*), $p < 0.01$ (**), $p < 0.001$ (***), and $p < 0.0001$ (****).

## Materials and data availability

All materials are available from the corresponding author upon request. High-content imaging datasets are deposited to the Image Data Resource (https://idr.openmicroscopy.org) under accession number idr0139. Data files for data retrieved from high content imaging analysis are provided as tables in Source Data files as indicated. Copies of the raw data in this manuscript are available for download from the King's Open Research Data System (KORDS) DOI: 10.18742/20408142. MATLAB script is available at https://github.com/IJAL98/NucActin, (copy archived at swh:1:rev:3418703575cab2e-9255d5957a4e85057651f3004, *Jayo, 2022*).

# Acknowledgements

We are grateful to Steve Lynham (King's College London Proteomics Facility) for assistance with proteomics experiments and to Dave Smith (AstraZeneca) for assistance with the small molecule compound library. The authors would also like to thank Angus Lamond (University of Dundee) and Hiroshi Kimura for generously providing plasmids for this study. The authors are also grateful to Simon Ameer-Beg and Conor Treacy (King's College London) for assistance with FLIM. CL and RJM are funded by the Medical Research Council UK (MR/R008264/1, to MP). MBD is funded by Cancer Research UK (City of London Centre Studentships award C7893/A31530, to MP).

# Additional information

### Competing interests

Samantha Peel: SP is affiliated with AstraZeneca; the author has no financial interests to declare. Adam Corrigan: AC is affiliated with AstraZeneca; the author has no financial interests to declare. Preeti Iyer: PI is affiliated with AstraZeneca; the author has no financial interests to declare. Jan Gettemans: JG is shareholder of the company Gulliver Biomed BV. JG declares that he has no non-financial competing interests. Maddy Parsons: Reviewing editor, eLife. The other authors declare that no competing interests exist.

## Funding

| Funder | Grant reference number | Author |
|---|---|---|
| Medical Research Council | MR/R008264/1 | Maddy Parsons |
| Cancer Research UK | C7893/A31530 | Maddy Parsons |

The funders had no role in study design, data collection and interpretation, or the decision to submit the work for publication.

## Author contributions

Campbell D Lawson, Conceptualization, Data curation, Formal analysis, Investigation, Methodology, Writing – original draft; Samantha Peel, Conceptualization, Data curation, Formal analysis, Supervision, Methodology, Writing – review and editing; Asier Jayo, Conceptualization, Data curation, Formal analysis, Investigation, Writing – review and editing; Adam Corrigan, Software, Supervision, Visualization, Methodology, Writing – review and editing; Preeti Iyer, Data curation, Software, Formal analysis, Writing – review and editing; Mabel Baxter Dalrymple, Formal analysis, Investigation, Methodology, Writing – review and editing; Richard J Marsh, Conceptualization, Software, Writing – review and editing; Susan Cox, Resources, Software, Supervision, Writing – review and editing; Isabel Van Audenhove, Resources, Validation, Investigation, Methodology, Writing – review and editing; Jan Gettemans, Resources, Formal analysis, Supervision, Funding acquisition, Methodology, Writing – review and editing; Maddy Parsons, Conceptualization, Resources, Data curation, Formal analysis, Supervision, Funding acquisition, Writing – original draft, Project administration

## Author ORCIDs

Campbell D Lawson [ID] http://orcid.org/0000-0001-5349-0638
Asier Jayo [ID] http://orcid.org/0000-0002-9899-8723
Maddy Parsons [ID] http://orcid.org/0000-0002-2021-8379

## Decision letter and Author response

Decision letter https://doi.org/10.7554/eLife.79283.sa1
Author response https://doi.org/10.7554/eLife.79283.sa2

## Additional files

### Supplementary files

• MDAR checklist

### Data availability

All materials are available from the corresponding author upon request. High content imaging datasets are deposited to the Image Data Resource (https://idr.openmicroscopy.org) under accession number idr0139. Data files for data retrieved from high content imaging analysis are provided as tables in figure supplement files as indicated. Copies of the raw data in this manuscript are available for download from the King's Open Research Data System (KORDS) https://doi.org/10.18742/20408142. MATLAB script is available at https://github.com/IJAL98/NucActin (copy archived at swh:1:rev:3418703575cab2e9255d5957a4e85057651f3004).

The following dataset was generated:

| Author(s) | Year | Dataset title | Dataset URL | Database and Identifier |
|---|---|---|---|---|
| Lawson P | 2022 | Nuclear fascin regulates cancer cell survival | https://doi.org/10.18742/20408142 | figshare, 10.18742/20408142 |

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
