## [Editor Report]

This paper significantly extends previous work suggesting a role for fascin in the nucleus, with the authors concluding that it contributes to multiple aspects of cancer cell regulation and behaviour. The authors used a combination of methodologies, including biochemistry, excellent cell and actin imaging, controlled nanobody-mediated targeting of fascin to the nucleus (when endogenous fascin has been suppressed), proteomics, cancer cell biological assays, and high-content phenotypic screening to identify potential regulators, and function, of nuclear fascin, and the consequences of maintaining too high levels of fascin in the nucleus. Fascin is an important protein in cancer cell behaviour, and this work provides novel information on dynamic active transport in and out of the nucleus, on its role in nuclear actin bundling, its binding to histone-H3, and its contribution to the DNA damage response (DDR; monitored by gammaH2AX foci accumulation), chromatin compaction, cell migration, and cancer cell invasion in an in vitro assay; moreover, dynamic regulation of nuclear fascin is important because too much triggers apoptosis.

---

## [Decision Letter]

**Decision letter after peer review:**

Thank you for submitting your article "Nuclear fascin regulates cancer cell survival" for consideration by *eLife*. Your article has been reviewed by 3 peer reviewers, including Margaret C Frame as Reviewing Editor and Reviewer #1, and the evaluation has been overseen by Jonathan Cooper as the Senior Editor. The following individuals involved in the review of your submission have agreed to reveal their identity: Chris Bakal (Reviewer #2); Adam Byron (Reviewer #3).

Recommendations for authors:

We would support publication after improvements as outlined below.

There are three main areas that would benefit from further data, statistics, and clarifications.

Can additional controls be provided to demonstrate evidence for direct interaction between Histone H3 and fascin using FRET? Minimally quantification of each fluor (GFP, cherry) when co-expressed and when individually expressed (to account for bleed through), a positive control, and negative control. Would a useful negative control to include for the FRET in Figure 2F be the mCh-Nb2-NES?

The role of fascin in the DDR warrants further exploration. Given the obvious variability in the data, are box and whisker plots the most appropriate (see specific points to authors, point 5)? Could the use of additional damaging agents, reporters, and live cell assays be considered to help in figuring out the role of fascin in the DDR?

Can the authors provide further validation of the mechanistic hypothesis that Haspin is promoting fascin-dependent nuclear actin polymerization, DNA damage, and decreased viability via effects on histone H3? Could the authors show clearer images of fascin relocalisation and actin polymerization? Are there experiments that could causally link nuclear fascin and actin to induction of DNA damage and/or demonstrate a role for histone H3 in this? What is being quantified in Figures 7G and H? Is this recruitment of γH2X to nuclear foci (under conditions of no other nuclear damaging agent)?

Details of analyses to include and consider:

1. The quantification of nuclear F-actin fibre organisation (Figure 1E) would benefit from some explanation in the Results section. We appreciate this is detailed in the methods section, but a brief high-level description of what is being quantified would help the reader interpret the results.

2. Could the authors provide some additional reporting information for the proteomics methods?

a. Version/release of the human database searched.

b. Variable and fixed modifications used in the database search.

c. Enzyme specificity and number of permitted missed cleavages set in the database search.

d. Minimum number of unique peptides specified in the database search.

e. Number of biological and/or technical replicates conducted.

f. Any normalisation of the MS data (e.g. to bait).

g. Any statistical analyses of the MS data performed (e.g. to threshold prioritised hits).

3. Further annotation/detail would improve the presentation of the MS data in Supplementary Table 1.

a. From where were the cellular component term headers derived?

b. What do the final four columns (Z-AC) represent?

c. Two hidden rows of data (32-33) should be unhidden from the table.

d. Column A suggests 238 proteins were identified, but only 59 are shown. Ideally, all identified proteins should be presented in a supplementary table, and prioritised hits (based on an appropriate set of criteria) can be presented in another supplementary table.

e. Protein names (column A) should be separated from the appended database-derived controlled vocabulary to improve clarity.

f. For discussion of the MS data, was fascin itself identified in the proteomics experiments?

4. Will the authors deposit the MS data in a suitable repository, such as those in the ProteomeXchange Consortium, to ensure accessibility and reusability of the data?

5. Some statistical analyses might be clarified.

a. Can the authors confirm what statistical test was used in Figure 3B?

b. For Figure 3F, did the authors test for a significant difference between fascin KD and fascin KD + NCS, which is stated to be unchanged in the text?

c. For experiments with more variable data (e.g. the DDR experiments), the authors should consider presenting charts with individual data points as a violin plot or as a histogram.

6. Why do the 0-μm images for the collagen invasion assay (Figure 4D) show much higher cell densities for the Nb2-NLS and fascin KD conditions compared to the other conditions? The image stacks look left-shifted for these two conditions. Are these z-stacks correctly positioned, and how was the "starting" slice determined representatively?

7. For Figure 7H, the correlation is said to be 'significant'. What statistical test was used here?

8. The final sentence of the legends for Figures1-4 should read min/mix (not "mix/max").

---

## [Author Response]

Recommendations for authors:We would support publication after improvements as outlined below.There are three main areas that would benefit from further data, statistics, and clarifications.Can additional controls be provided to demonstrate evidence for direct interaction between Histone H3 and fascin using FRET? Minimally quantification of each fluor (GFP, cherry) when co-expressed and when individually expressed (to account for bleed through), a positive control, and negative control. Would a useful negative control to include for the FRET in Figure 2F be the mCh-Nb2-NES?

We thank the reviewers for this comment. We would like to clarify that we used Fluorescence Lifetime Imaging Microscopy (FLIM) to measure FRET in our manuscript. FLIM (unlike other ratiometric or sensitised emission techniques) measure the inherent fluorescence lifetime of the donor molecule (in this instance, GFP) and as such, is entirely independent from fluorophore concentrations and is not subject to bleed through. All FRET efficiencies from FLIM datasets are calculated as a function of GFP alone donor lifetime (~2.3ns, an example of which we showed in our original submission Figure 2F). However, as requested we have included two additional control datasets in our revised manuscript: GFP-H3 + mCherry alone and as very helpfully suggested by the reviewers, GFP-H3 + mCh-Nb2-NES. The data from these experiments shows no FRET occurring between either control donor: acceptor pair and is included in Figure 2G of our revised manuscript. This data further supports the conclusions from our experiments that nuclear fascin directly associates with Histone H3.

The role of fascin in the DDR warrants further exploration. Given the obvious variability in the data, are box and whisker plots the most appropriate (see specific points to authors, point 5)? Could the use of additional damaging agents, reporters, and live cell assays be considered to help in figuring out the role of fascin in the DDR?Can the authors provide further validation of the mechanistic hypothesis that Haspin is promoting fascin-dependent nuclear actin polymerization, DNA damage, and decreased viability via effects on histone H3? Could the authors show clearer images of fascin relocalisation and actin polymerization? Are there experiments that could causally link nuclear fascin and actin to induction of DNA damage and/or demonstrate a role for histone H3 in this?

We thank the reviewers for these comments and thoughts. We agree that the precise mechanisms by which fascin-Histone H3 binding and nuclear F-actin lead to DDR regulation are not fully delineated in our study, although we would note that unpicking this entire mechanistic pathway would likely need to be the focus of a separate study. However, we have performed 3 additional sets of experiments to try and further validate our proposed model.

Firstly, we expressed WT, S39A (constitutive actin bundling) or S39D (non-actin bundling) RFP-fascin in fascin KD cells and co-expressed the fascin GFP-Nb2-NLS nanobody, followed by GFP-Nb2-NLS immunoprecipitation and re-probing for Histone H3. These experiments were designed to test whether nuclear fascin:F-actin binding is mutually exclusive with fascin:Histone H3 binding (as we hypothesise). Indeed, our data demonstrate that nuclear fascin-S39A does not associate with Histone H3, whereas WT and fascin-S39D do, supporting the idea that fascin associates with (likely de-phosphorylated as our other data suggests) Histone H3 when not bound to F-actin. This data is included as Figure 2H of our revised manuscript.

Secondly, we have performed live imaging experiments of fascin KD cells expressing mScarlet-fascin, GFP-Histone H3 and iRFP-nAC +/- haspin inhibitor to gain further insight into temporal fascin-dependent changes to nuclear F-actin and histone organisation and dependence upon Histone H3 phosphorylation. The resulting data demonstrate that in untreated cells in interphase, nuclear fascin levels are low, nuclear F-actin structures are absent or limited and Histone H3 appears as punctate structures (similar to the pHistone H3 images we show in Figure 7F). However, addition of the haspin inhibitor leads to higher nuclear fascin (as we already demonstrated using multiple approaches), highly bundled, stable F-actin structures in the nucleus that appear to result in scaffolding or reorganisation of Histone H3. This data is included as Figure 7G of the revised manuscript. We note that we have spent a considerable amount of time trying to optimise conditions to perform STORM super-resolution imaging of nuclear fascin and actin (along with other targets of interest) to better understand the nature of organisation of these molecules. However, retrieving super-resolved information in multiple channels from nuclear compartments has proven to be exceptionally challenging and whilst we plan to try and overcome the technical hurdles to undertake such analysis, we anticipate this will likely need considerable additional time commitment to achieve.

Finally, as suggested we have included cisplatin treatment of cells as an alternative mechanistic driver of DDR (and widely used as standard of care chemotherapy in the clinic). The data from these experiments mirror those seen with NCS treatment and further confirms that efficient induction of DDR requires fascin. This data is now included as Figure 3C of our revised manuscript.

We note we have already performed fixed time course experiments using endogenous markers of DDR to demonstrate a requirement for nuclear fascin in the temporal induction and resolution of damage (shown in Figure 3B). Whilst we appreciate that live imaging of DDR might provide additional insight into this process, published fluorescent reporters of this process largely rely on overexpression which comes with caveats of disturbing the balance of DDR response, likely leading to artefacts (as noted by the field in recent reviews on this matter, e.g.: PMID: 34589495). Given the limited reliable and interpretable tools available for such analysis, and the rather limited additional insight we feel this might bring, we did not feel such experiments would meaningfully add to our current manuscript. As we outline in the discussion, the exact molecular mechanisms driving efficient DDR upon fascin disengagement from Histone H3, and assembly of bundled nuclear F-actin remain undefined. However, given the previous studies indicating a role for F-actin in DDR, and our own data presented here showing that fascin-dependent nuclear F-actin is required for efficient DDR and chromatin organisation, we believe our data supports fascin-dependent nuclear F-actin acting as a scaffold for DDR protein recruitment. Further studies will be required to define how this occurs through precise alterations to (for example) chromatin structure or accessibility.

We have re-plotted the DDR data in Figure 3B and the FRET data in Figure 3F as violin plots with all datapoints shown as requested (noting DDR data in Figure 7 was already plotted in this way).

What is being quantified in Figures 7G and H? Is this recruitment of γH2X to nuclear foci (under conditions of no other nuclear damaging agent)?

We apologise if this was unclear. Figures 7G and 7H (now Figures 7H and 7I, respectively) show levels of γH2AX recruited to nuclear DNA damage foci in cells treated only with DMSO or the Haspin inhibitor (7H) and this data was then plotted as a function of nuclear fascin from those images (7I, nuclear fascin on x-axis and γH2AX on y-axis), showing correlation between nuclear fascin and DDR. The data indicates that inhibition of Haspin leads to increased γH2AX levels (as another recent study has also shown; PMID:31882401) and that this correlates with enhanced retention of nuclear fascin.

Details of analyses to include and consider:1. The quantification of nuclear F-actin fibre organisation (Figure 1E) would benefit from some explanation in the Results section. We appreciate this is detailed in the methods section, but a brief high-level description of what is being quantified would help the reader interpret the results.

We thank the reviewers for this suggestion and have added additional explanation of this analysis approach into the results text for Figure 1.

2. Could the authors provide some additional reporting information for the proteomics methods?a. Version/release of the human database searched.b. Variable and fixed modifications used in the database search.c. Enzyme specificity and number of permitted missed cleavages set in the database search.d. Minimum number of unique peptides specified in the database search.e. Number of biological and/or technical replicates conducted.f. Any normalisation of the MS data (e.g. to bait).g. Any statistical analyses of the MS data performed (e.g. to threshold prioritised hits).

We apologise for these omissions; the information requested in a-e has now been included in the revised Figure 2-Source Data 2 file as a separate tab and referred to in the methods section. We did not perform normalisation (values provided in Supplementary table 1 are unique peptide counts) and we conducted n=2 samples per IP (both datasets provided in the Supp table), so statistical analysis was not possible. However (and see below for further discussion), we would note this was not a full proteome analysis, but (as we describe in the methods section) was rather conducted on excised bands from silver-stained immunoprecipitations where differential levels of proteins were observable between NES and NLS Nb IPs. The experiment was not performed to statistically identify full proteome changes, rather to define indicative targets that were consistently different between NLS and NES samples and thus may represent differential nuclear fascin-binding proteins. Prioritisation was performed on this basis, and we have clarified this point in the revised methods section. Validation of identified targets was used to ascertain specificity.

3. Further annotation/detail would improve the presentation of the MS data in Supplementary Table 1.a. From where were the cellular component term headers derived?b. What do the final four columns (Z-AC) represent?c. Two hidden rows of data (32-33) should be unhidden from the table.d. Column A suggests 238 proteins were identified, but only 59 are shown. Ideally, all identified proteins should be presented in a supplementary table, and prioritised hits (based on an appropriate set of criteria) can be presented in another supplementary table.e. Protein names (column A) should be separated from the appended database-derived controlled vocabulary to improve clarity.f. For discussion of the MS data, was fascin itself identified in the proteomics experiments?

We apologise for the missing information requested in points a-d, which is now provided in the revised Figure 2-Source Data 2 file. We have fixed moved protein classifiers to the far right of the spreadsheets to focus on the protein names as suggested. As per our above explanation, fascin was not identified in the datasets as the analysed samples were excised bands from silver-stained gels not the full proteomes.

4. Will the authors deposit the MS data in a suitable repository, such as those in the ProteomeXchange Consortium, to ensure accessibility and reusability of the data?

As per our above explanations, this was not a full proteome analysis and as such, we did not feel that this represents a useful dataset for community re-use given it was performed on selected regions of silver-stained gels. However, as requested, the full dataset is provided in the revised Figure 2-Source Data 2 file (as a separate tab) such that all identified proteins are present.

5. Some statistical analyses might be clarified.a. Can the authors confirm what statistical test was used in Figure 3B?b. For Figure 3F, did the authors test for a significant difference between fascin KD and fascin KD + NCS, which is stated to be unchanged in the text?c. For experiments with more variable data (e.g. the DDR experiments), the authors should consider presenting charts with individual data points as a violin plot or as a histogram.

Apologies if these points were unclear. Data in Figure 3B were analysed using 2-way ANOVA followed by Dunnet’s multiple comparison test (as stated in the methods section); we present what we deemed to be the most biologically relevant statistics on this graph. We did perform statistical analysis on fascinKD+/-NCS in Figure 3F and found no significant difference as stated in the results text (p=0.1549). We have re-plotted the DDR data in Figure 3B and the FRET data in Figure 3F as violin plots with all datapoints shown as suggested (noting DDR data in Figure 7 was already plotted in this way).

6. Why do the 0-μm images for the collagen invasion assay (Figure 4D) show much higher cell densities for the Nb2-NLS and fascin KD conditions compared to the other conditions? The image stacks look left-shifted for these two conditions. Are these z-stacks correctly positioned, and how was the "starting" slice determined representatively?

The 0 mm images for Nb2-NLS and Fascin KD cells do indeed show higher cell densities as these cells invade significantly less than other cells, meaning more cells remain at the base of the inverted invasion assay. As we stated in the methods, the 0 mm images are taken as the first Z-stack at which more than 5 whole nuclei are visible using identical imaging parameters, and all subsequent invasive depth measurements calculated from that point. We have clarified both points (in the results and methods text respectively) in the revised manuscript.

7. For Figure 7H, the correlation is said to be 'significant'. What statistical test was used here?

Apologies for the omission; we used Pearson’s correlation coefficient analysis in GraphPad Prism to test this data, used to interpret the confidence interval of r and the P value testing the null hypothesis that there is no correlation between the two variables and any correlation observed is a consequence of random sampling. We have added this into the methods section of the revised manuscript.

8. The final sentence of the legends for Figures1-4 should read min/mix (not "mix/max").

We apologise for these errors, which have now been corrected in the revised manuscript (noting the figure legends have now been moved to the main manuscript document in-line with *eLife* formatting requirements for resubmissions).